# Review of Layered Transition Metal Oxide Materials for Cathodes in Sodium-Ion Batteries

**DOI:** 10.3390/mi16020137

**Published:** 2025-01-24

**Authors:** Mehdi Ahangari, Meng Zhou, Hongmei Luo

**Affiliations:** Department of Chemical and Materials Engineering, New Mexico State University, Las Cruces, NM 88003, USA

**Keywords:** sodium-ion batteries, layered transition metal oxides, cathode materials

## Abstract

The growing interest in sodium-ion batteries (SIBs) is driven by scarcity and the rising costs of lithium, coupled with the urgent need for scalable and sustainable energy storage solutions. Among various cathode materials, layered transition metal oxides have emerged as promising candidates due to their structural similarity to lithium-ion battery (LIB) counterparts and their potential to deliver high energy density at reduced costs. However, significant challenges remain, including limited capacity at high charge/discharge rates and structural instability during extended cycling. Addressing these issues is critical for advancing SIB technology toward industrial applications, particularly for large-scale energy storage systems. This review provides a comprehensive analysis of layered sodium transition metal oxides, focusing on their structural properties, electrochemical performance, and degradation mechanisms. Special attention is given to the intrinsic and extrinsic factors contributing to their instability, such as structural phase transitions, and cationic/anionic redox behavior. Additionally, recent advancements in material design strategies, including doping, surface modifications, and composite formation, are discussed to highlight the progress toward enhancing the stability and performance of these materials. This work aims to bridge the knowledge gaps and inspire further innovations in the development of high-performance cathodes for sodium-ion batteries.

## 1. Introduction

The urgent global need for clean, affordable, and sustainable energy storage systems has become increasingly critical due to the escalating effects of climate change, the rapid depletion of fossil fuel resources, and rising environmental pollution. To address these challenges, the adoption of advanced technologies for energy storage and conversion has gained immense importance. Over the past few decades, renewable energy sources such as wind, solar, nuclear, and wave energy have emerged as promising alternatives. However, the intermittent nature of these energy sources, influenced by factors like time, weather, and location, has limited their widespread application, underscoring the need for efficient energy storage systems [1,2,3,4].

Electrochemical energy storage devices, including batteries, fuel cells, and supercapacitors, are among the most effective technologies for storing electricity and powering modern devices. Among these, rechargeable (secondary) batteries stand out for their superior energy and power density compared to fuel cells and supercapacitors. Since their commercialization by Sony in the 1990s, lithium-ion batteries (LIBs) have become a cornerstone of modern society, transforming how we interact with technology by enabling wireless and portable solutions. LIBs have dominated battery technology for decades, powering devices such as laptops, smartphones, and even electric and hybrid vehicles, significantly reducing dependence on fossil fuels [5,6,7,8].

Lithium’s unique properties, including high voltage, energy density, and specific power compared to other elements (Figure 1a) make it an ideal material for battery applications, particularly in the automotive industry. However, the surging demand for LIBs has led to a steep rise in the cost of lithium-based precursors like lithium carbonate (Li_2_CO_3_) over the past decade (Figure 1b). Additionally, the geographically constrained lithium reserves, predominantly located in Latin America and Australia, have raised concerns about supply security and cost, especially for large-scale applications. These challenges highlight the need for alternative energy storage solutions that are cost-effective, widely available, and sustainable (Figure 1c) [9,10,11].

Sodium-ion batteries (SIBs) share a similar architecture and working principles with LIBs (Figure 2), making them a promising alternative. They are environmentally benign, cost-effective, and abundant, addressing several limitations associated with LIBs. Research into SIBs began in the early 1980s, but the successful commercialization of LIBs in the 1990s shifted focus away from sodium-based systems [12]. Table 1 provides a general comparison between sodium and lithium batteries, highlighting their key characteristics and differences.

In addition to SIBs, solid-state sodium batteries are emerging as a promising candidate for the future of large-scale energy storage applications. The organic electrolytes currently used in SIBs—such as propylene carbonate, ethylene carbonate, and dimethyl carbonate—struggle to meet the performance requirements of both cathodes and anodes, particularly in achieving high electrochemical performance. Moreover, the flammability of these liquid electrolytes raises significant safety concerns, as seen in lithium-ion batteries, particularly in large battery systems for vehicles and grid applications. Solid electrolytes offer a compelling solution by effectively mitigating the risks of short circuits caused by sodium dendrites and eliminating the hazards associated with flammable organic electrolytes. The use of solid-state batteries, which leverage the inherent non-flammability of solid electrolytes, is anticipated to address these safety issues comprehensively [13,14].

Several types of solid electrolytes have been explored for rechargeable sodium batteries, including Na-ion conductive ceramics, polymers, and ceramic–polymer composites. Among these, organic polymer electrolytes—created by dissolving sodium salts into a polymer matrix—have garnered significant attention. These materials offer high flexibility, excellent thermal stability, and the ability to form intimate contact with both electrodes and inorganic solid electrolytes. With moderate ionic conductivity (>10^−4^ S cm^−1^ at room temperature), polymer-based solid electrolytes remain one of the most widely investigated options for solid-state sodium batteries [15,16,17,18].

Sodium is one of the most abundant elements in the Earth’s crust, with a Clarke number of 2.64 (Figure 3a). It exists in various forms such as sodium carbonate (Na_2_CO_3_), sodium chloride (NaCl), and sodium sulfate (Na_2_SO_4_), commonly found in minerals and brines. Sodium also exhibits an electrochemical redox potential close to that of lithium (−2.71 V for sodium versus −3.04 V for lithium, both measured against the standard hydrogen electrode), making it suitable for battery applications [19,20].

The abundance of sodium resources makes it a highly accessible material, with significantly lower costs compared to lithium. For example, the cost of trona, a key source of sodium carbonate, ranges between USD 135 and USD 165 per ton, whereas lithium carbonate, a precursor material for LIBs, cost approximately USD 5000 per ton in 2010. This stark contrast in resource availability and cost highlights sodium as a viable alternative for scalable and cost-effective energy storage systems (Figure 3b,c). Therefore, the cost per kilowatt-hour (USD/kWh) for SIBs can be significantly reduced due to the absence of expensive transition metals like cobalt and nickel in the cathode materials.

Additionally, aluminum can replace copper as the current collector for the anode because sodium does not react with aluminum. This allows the aluminum current collector to remain stable in the electrolyte at 0 V, unlike in LIBs, where lithium and copper form a binary alloy at low potentials (below 0.1 V vs. Li/Li^+^).

However, sodium’s larger ionic radius (1.02 A° compared to 0.76 A° for lithium) and heavier atomic mass (23 g mol^−1^ vs. 7 g mol^−1^ for lithium) contribute to slower ion kinetics during the (de)sodiation process and lower gravimetric capacity (1165 mAh g^−1^ compared to 3829 mAh g^−1^). This leads to structural instability in the host materials, increased heat generation from exothermic reactions, and a lower theoretical energy density. While high energy density is not a primary requirement for stationary, in large-scale energy storage systems—such as those paired with solar panels where cost efficiency is prioritized—these issues still negatively affect the performance of SIBs [23,24,25].

Moreover, the solid electrolyte interphase (SEI) formed on the electrodes in SIBs is less stable due to sodium’s lower Lewis acidity, leading to a high solubility of the SEI components. This results in incomplete coverage of the electrode surface, promoting side reactions and excessive heat generation. These factors highlight that the electrochemical behavior of SIBs cannot be directly extrapolated from LIBs. To overcome these challenges and improve performance, comprehensive studies are essential to understanding the unique electrochemical processes and underlying mechanisms of SIBs.

Cathode materials reported for SIBs include layered oxides, polyanions, organic compounds, and Prussian blue and its analogues. Despite their potential, these materials face persistent challenges such as low ionic conductivity and thermal instability, which limit their performance. Nonaqueous organic liquid electrolytes are predominantly used in SIBs due to their wide electrochemical stability window, high ionic conductivity, and efficient mass transfer at the electrolyte–electrode interface. However, they also present safety concerns, including flammability and potential for thermal runaway.

Identifying a cathode material with superior electrochemical properties—such as high capacity, long cycle life, and optimal operating voltage—remains a top priority for advancing SIB technology. Currently, cathode materials in SIBs typically exhibit specific capacities ranging from 120 to 200 mA h g^−1^, with operating potentials between 2.6 and 3.2 V. These values are considerably lower than the oxidation potential of electrolytes, which exceeds 4.0 V. In contrast, key anode materials offer specific capacities of 300–600 mA h g^−1^, with operating potentials between 0.1 and 0.6 V, which are close to the potential for sodium plating. Consequently, cathode materials not only represent a larger mass fraction in SIBs, but they also offer more potential for enhancing the overall voltage of the system. Therefore, cathodes play a critical role in determining the cost, safety, specific energy, cycle life, and specific power of SIBs. Unfortunately, the fundamental differences in intercalation chemistry between lithium and sodium often result in the underperformance of sodium-based analogues of lithium-based compounds. These differences underscore the need for tailored materials and innovative design strategies to optimize the performance of SIB cathodes [26,27,28,29,30].

The α-NaFeO_2_ (R-3m) structured materials hold a pivotal role in the development of LIBs and serve as a foundation for similar materials in SIBs. These materials feature a layered structure composed of slabs of edge-sharing MO_6_ octahedra, with alkali ions intercalated between the layers [31]. Similar to lithium-based transition metal oxides, such as LiCoO_2_ and Li(Ni, Mn)O_2_, sodium transition metal oxides (Na_x_TMO_2_, where TM = transition metals and 0.67 ≤ x ≤ 1) are commonly used as positive electrodes in SIBs [32,33,34].

Typical sodium layered oxides can be synthesized in O3-, P2-, and P3-phases, which are determined by the surrounding Na environment and the number of unique oxide layer packings. The notation O3, P3, and P2 describe the coordination environment and stacking order: O indicates that sodium ions occupy octahedral (O) sites, while P signifies sodium ions in trigonal prismatic (P) sites. The numbers 3 and 2 represent the number of edge-sharing MO_6_ octahedra in the oxygen stacking sequence, arranged in ABCABC or ABBA patterns, respectively [35]. The O3 phase typically occurs when 0.7 ≤ x ≤ 1. The oxide layer stacking follows the ABCABC pattern, where all Na ions share one edge and one face with adjacent octahedra. When 0.6 < x < 0.7, the P2 phase is known to occur, which features oxide layer stacking that follows the ABBA pattern, with Na ions sharing either entirely edge or entirely face positions with adjacent octahedra. P3 typically occurs when x ~ 0.5, the oxide layer stacking in this phase follows the ABBCCA pattern, where Na ions share one face with a transition metal octahedron (MO_6_) and three edges with three adjacent MO_6_ octahedra, as illustrated in Figure 4 [36]. The O3 phases belong to the R-3m space group, whereas P2 phases crystallize in the P63/mmc space group. Structural distortions in these phases, such as monoclinic (C2/m) or orthorhombic (Cmcm) modifications, are denoted by a prime symbol (for example, O3′ or O′3) [37,38].

The extraction of alkali ions during battery operation creates vacancies in the Li or Na layers, which can facilitate the migration of TMs into these sites. This migration leads to irreversible structural degradation and poor electrochemical performance in layered oxides. The phenomenon of TM migration has been extensively studied in LIBs cathode materials. For instance, Wang et al. [39] observed the migration of Ni and Co ions in LiNi_1/3_Mn_1/3_Co_1/3_O_2_ and attributed it to structural distortions and the formation of irreversible phases.

Both experimental observations and theoretical calculations indicate that distortions in the MO_6_ octahedron play a critical role in promoting TM migration. These distortions may arise from TM oxidation or oxygen oxidation during cycling [40,41]. A key characteristic of these oxidation processes is the alteration in TM-O bond lengths, which induces strain within the MO_6_ octahedra or the O-TM-O slabs. This strain serves as a driving force for TM migration, ultimately impacting the structural integrity and electrochemical stability of the cathode material.

The cycling life of cathode materials in SIBs is primarily influenced by the structural stability of the materials, as well as the electrochemical stability of the compatible electrolytes. Electrochemical behavior is significantly influenced by the structure of the phase, not only because of the amount of Na in the pristine state but also due to the stability of each layer. The kinetics of sodium insertion and extraction are affected by the surrounding environment of Na within the layered structure, which impacts the overall performance of the material. The stability of each layer plays a crucial role in maintaining structural integrity during cycling, while the arrangement and interaction of Na ions within the structure influence the efficiency of ion transport and the electrochemical response. This review focuses on the phase transitions, structural stability, and electrochemical characteristics of transition metal layered oxide materials, as well as various strategies reported to enhance the stability of these materials for SIBs.

**Figure 4 micromachines-16-00137-f004:**
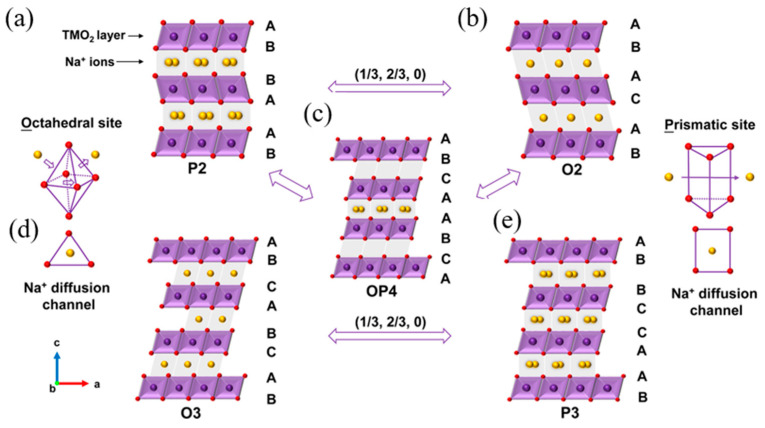
Schematic illustration of the typical crystal structures of Na_x_TMO_2_, showing (**a**) P2, (**b**) O2, (**c**) OP4, (**d**) O3, and (**e**) P3 phases. Purple spheres represent transition metals, yellow spheres denote Na^+^ ions, and red spheres indicate O^2−^ ions. Reprinted from Ref. [42] with permission from Wiley.

## 2. Materials Classification

### 2.1. NaCrO_2_

Among Na_x_MO_2_ compounds (0 < x ≤ 1, M = Ni, Co, Mn, Fe, V, and Cr), O3-NaCrO_2_ has garnered significant attention as a promising cathode material for sodium-ion batteries. It delivers a discharge capacity of 120 mAh g^−1^ with a desirable voltage plateau at ~3 V (vs. Na/Na^+^) and exhibits excellent thermal stability compared to LiCoO_2_ and LiFePO_4_. This stability is attributed to the robust chemical stability of Cr^4+^ within CrO_2_ slabs at octahedral sites, combined with high-rate capability and excellent coulombic efficiency [43,44,45].

Interestingly, while LiCrO_2_, which shares a layered rock salt structure with LiCoO_2_ and LiNiO_2_, is electrochemically inactive, NaCrO_2_ demonstrates high capacity. This discrepancy stems from differences in interstitial spaces between the two compounds. In LiCrO_2_, the irreversible migration of Cr ions into tetrahedral sites during lithium de-intercalation renders the material inactive after the first charge. In contrast, NaCrO_2_ avoids this migration when x ≤ 0.5 due to its larger tetrahedral sites, although further sodium removal (x > 0.5) leads to gradual inactivation [46,47]. Structurally, both LiCrO_2_ and NaCrO_2_ feature Cr in the trivalent state with a d^3^ electronic configuration (t_2g_^3^, e_g_^0^) under octahedral coordination. Upon oxidation, Cr(III) forms the unstable Cr(IV) (d^2^), which disproportionate into Cr(III) (d^3^) and Cr(VI) (d^0^) due to their higher stability. Cr(VI) typically adopts a tetrahedral coordination with four oxygen atoms, as seen in Cr(VI)O_4_^2−^ of Li_2_CrO_4_·2H_2_O. The dimensional match between the interstitial tetrahedron and Cr(VI)O_4_^2−^ plays a crucial role in structural stability [48].

For LiCrO_2_, the interstitial tetrahedron with an O-O bond length of 2.89 A° and a height of 2.56 A° closely aligns with the Cr(VI)O_4_^2−^ tetrahedron. However, in NaCrO_2_, the interstitial tetrahedron within the NaO_2_ slab exhibits an O-O bond length of 2.97 A° and a height of 3.15 A°, better accommodating the Cr(VI)O_4_ structure. Consequently, the bond length mismatch in NaCrO_2_ prevents irreversible Cr migration, unlike in LiCrO_2_ (Figure 5). Magnetic measurements of chemically deintercalated Na_1-x_CrO_2_ confirm the presence of Cr(IV) during sodium deintercalation.

In NaCrO_2_, oxygen atoms at the 6c sites are stacked along the c-axis, with Na and Cr ions alternately distributed on both sides of the oxygen layers at the 3a and 3b octahedral sites. This layered structure enables two-dimensional Na^+^ ion transport between CrO_2_ slabs, enhancing ionic mobility. However, consecutive phase transitions during cycling—from O3 to O′3 and eventually P3—pose significant challenges.

To date, most NaCrO_2_ materials have been synthesized via solid-state reactions, which suffer from capacity decay and poor power performance due to irreversible phase transitions, low Na^+^ diffusivity, and side reactions at the electrode–electrolyte interface [7]. Addressing these issues is critical for realizing the full potential of NaCrO_2_ as a cathode material in sodium-ion batteries.

The initial charge and discharge curves for NaCrO_2_ and LiCrO_2_ are shown in Figure 6a,b, respectively. LiCrO_2_, upon electrochemical oxidation to 4.5 V vs. Li, demonstrates a capacity of 55 mAh g^−1^, corresponding to the formation of Li_0.81_CrO_2_. This capacity is consistent with the charge compensation mechanism driven by the increased valence of Cr ions during lithium extraction. A slight reversible capacity is observed in the potential range of 3.0–4.5 V; however, subsequent cycles reveal a drastic drop in capacity to less than 10 mAh g^−1^, signifying the absence of reversible lithium intercalation. This behavior is attributed to the so-called ‘freeze effect’ of Cr^6+^ ions migrating into tetrahedral sites post-delithiation, as suggested by Komaba et al. [49,50,51].

In the case of NaCrO_2_ synthesized by traditional solid-state reaction of stoichiometric Cr_2_O_3_ and Na_2_CO_3_ at 900 °C in Ar environment, sodium deintercalation occurs electrochemically, exhibiting multiple plateaus (Figure 6b) corresponding to phase transitions: hexagonal O3 → monoclinic O3 → monoclinic P3 [52]. However, when cycled to a high cutoff voltage of 4.5 V, the intercalation process becomes nearly irreversible, resulting in a limited reversible capacity of 26 mAh g^−1^ (Figure 6c). This suggests that complete sodium deintercalation induces an irreversible structural change due to the destabilization of the deintercalated phase, which lacks the sodium pillars necessary to maintain the structural integrity of CrO_2_ slabs.

At a lower cutoff voltage of 3.6 V, the charge–discharge curves reveal highly reversible behavior with a capacity of 104 mAh g^−1^. This highlights that extracting more than 50% of sodium ions from NaCrO_2_ disrupts crystallinity, transitioning the material into a highly disordered and irreversible phase. The irreversible order-to-disorder transition during deep charging is attributed to Cr migration into vacant Na layers, driven by the disproportionation reaction of Cr(IV) at high potentials (3Cr^4+^ → 2 Cr^3+^ + Cr^6+^). The smaller ionic radius of Cr^6+^ and its preference for tetrahedral coordination facilitate this migration. Importantly, this phenomenon occurs exclusively in O-like phases, as P-like phases lack the necessary tetrahedral sites in their sodium-ion layers.

The disordered bulk structure is believed to comprise a mixture of O3-like Na_δ_CrO_2_ (δ ~ 0) and rock salt CrO_2_, as suggested by prior investigations [43]. These structural transformations underline the importance of optimizing charge cutoff voltages to mitigate irreversible phase changes and enhance the cycling stability of NaCrO_2_ cathodes.

Figure 7a,b depict the cycling performance of a Na/NaCrO_2_ cell. The distinct and repeatable plateaus observed in the charge and discharge curves demonstrate the reversible nature of Na^+^ intercalation within the NaCrO_2_ electrode [48].

Zhou et al. [51] provided a comprehensive analysis of the reaction mechanism and real-time structural evolution of the layered NaCrO_2_ cathode material during Na extraction. Utilizing synchrotron-based in situ XRD (Figure 8a), the structural changes in NaCrO_2_ were examined under a current density of C/12 and a cutoff voltage of 3.6 V.

At the onset of charging, the (003) diffraction peak shifted progressively to lower two-theta angles, indicative of a solid solution reaction accompanied by expansion along the c-axis. When x = 0.08, a new (003) peak emerged to the left of the original peak, and its intensity increased with continued charging. This indicated the formation of a distinct phase. In the range 0.08 < x < 0.2, a two-phase coexistence region was observed, with the intensity of the secondary (003) peak increasing at the expense of the first, while its position continued to shift to lower angles. By x = 0.25, the second phase dominated completely, with the original phase disappearing.

As the charge progressed to x = 0.3, another (003) peak appeared, and the second phase’s peak broadened, marking the formation of a third phase. By x = 0.4, only the third phase remained, and its (003) peak shifted further, albeit to a lesser extent than the second phase. This sequence of transitions demonstrated the systematic structural evolution during Na extraction.

Additional insights were obtained by analyzing the (110) peak, initially located at 62.4°, which reflected changes in the a-b lattice plane. As charging advanced, the sharp (110) peak diminished in intensity and broadened, indicating lattice distortion in the a-b plane. Concurrently, its shift to higher two-theta angles suggested a contraction in the a and b lattice parameters.

Based on the observed Bragg peak variations, the structural evolution of NaCrO_2_ involved phase transitions among three solid-solution phases, interspersed with two two-phase coexistence regions. Initially, the material adopted an O3-type rhombohedral phase (O3_R_). After x = 0.08, the second phase emerged, as evidenced by the splitting of the rhombohedral (012) peak, signaling a transition to a monoclinic phase. By x = 0.25, the second phase, identified as the monoclinic Na_0.74_CrO_2_ phase (O3_M_), dominated while maintaining O3 symmetry, albeit with distorted a-b planes and ABCABC oxygen atom stacking.

As x increased from 0.3 to 0.48, the third phase grew to dominate, and its Bragg reflections corresponded to the Na_0.52_CrO_2_ phase, identified as a P3-type monoclinic phase (P3_M_) with ABBCCA oxygen atom stacking. These findings highlight the complexity of the structural transformations occurring in NaCrO_2_ during electrochemical charging and their implications for its electrochemical performance.

The evolution of the a, b, and c lattice parameters during the initial charge of the NaCrO_2_ electrode, as derived from in situ XRD analysis, is presented in Figure 8b. During the charging process, the c-axis parameter exhibited an increase, which is attributed to the enhanced electrostatic repulsion between the MO_2_ layers as sodium ions were removed. These sodium ions act as a structural “glue” between the MO_2_ layers, and their extraction leads to an expansion of the interlayer spacing.

In contrast, the a and b lattice parameters decreased, reflecting a contraction of the CrO_6_ octahedra. This contraction is a direct consequence of the oxidation of Cr^3+^ to higher oxidation states during charging, resulting in shorter bond lengths and a more compact structure within the CrO_2_ slabs. These lattice parameter changes collectively highlight the structural adjustments occurring in NaCrO_2_ during sodium extraction.

The sodium extraction mechanism for the NaCrO_2_ electrode during the charging process, as elucidated through the authors’ experiments, can be summarized as follows:

Open Circuit Voltage (OCV) to 3.05 V:NaCrO_2_ (O3_R_) → Na_0.75_CrO_2_ (O3_M_) + 0.25 Na^+^ + 0.25 e^−^
(1)

3.05 V to 3.60 V:Na_0.75_CrO_2_ (O3_M_) → Na_0.5_CrO_2_ (P3_M_) + 0.25 Na^+^ + 0.25 e^−^
(2)

From the OCV to approximately 3.05 V, the O3_R_-type NaCrO_2_ undergoes a two-phase transformation into O3_M_-type Na_0.75_CrO_2_ (Reaction 1), facilitated by transitions through two solid-solution regions. Within each region, the ‘a’ and ‘b’ lattice parameters decrease, reflecting the contraction of the CrO_6_ octahedra, while the ‘c’ lattice parameter increases due to the reduced electrostatic interactions between the MO_2_ layers as sodium ions are extracted. This structural evolution is accompanied by a lattice distortion, transitioning from a rhombohedral to a monoclinic phase. During this process, Cr ions remain confined to octahedral sites, while Na ions migrate from octahedral to pseudo-tetrahedral sites.

Between 3.05 V and 3.60 V, the O3_M_-type Na_0.75_CrO_2_ undergoes another two-phase transformation into the P3_M_-type Na_0.5_CrO_2_ (Reaction 2), also involving two solid-solution regions. In this range, the Cr ions retain their octahedral coordination but decrease in size due to further oxidation. Simultaneously, Na ions transition from pseudo-tetrahedral to prismatic sites. These phase transitions are characterized by the gliding of oxygen planes, which is directly associated with the progressive removal of sodium ions.

Bo et al. [53] explored the structural evolution of NaCrO_2_ under a current density of C/50 up to 4 V, corroborating the structural changes observed by Zhou et al. [51] up to 3.6 V. As illustrated in Figure 9a, further sodium extraction beyond 3.8 V, corresponding to a Na_0.38_CrO_2_ composition, marks the onset of irreversible electrochemical behavior. This is evidenced by the appearance of diffraction peaks associated with new phase(s), notably a broad peak around 8°. In Figure 9b, these unidentified phases are denoted with an ‘X’, highlighting their ambiguous nature.

The transition from the P′3 phase to the unidentified ‘X’ phase progresses as the cutoff voltage of 4 V is approached, resulting in a system containing approximately 0.04 Na, as inferred from the capacity measurements. The phase transition near Na_0.4_CrO_2_ (around 3.7 V) and the significant decrease in discharge capacity observed beyond 3.8 V imply that this structural transformation is a key contributor to the capacity fading of NaCrO_2_.

In addition to phase transition characterization techniques, cyclic voltammetry (CV) was employed to investigate the structural changes occurring at both reversible and irreversible voltage ranges. Jakobsen et al. [54] conducted CV experiments in the reversible potential range of 2.5–3.7 V, as depicted in Figure 10a. During the initial charge, two distinct oxidation peaks were observed at 3.07 V (with a shoulder at 3.13 V) and 3.33 V, with corresponding reduction peaks at 2.83 V and 3.28 V during the first discharge. With repeated cycling, the potential hysteresis of the lowest potential event diminished, and after five cycles, the peaks shifted to 3.04 V and 2.87 V. Additionally, the reduction peak revealed a shoulder in the range of 2.9–2.95 V, analogous to the shoulder at 3.13 V during the charging process. Importantly, this shoulder is not associated with irreversibility, as this potential range is characterized by excellent cycling reversibility.

The authors attribute these features to several crystalline-to-crystalline structural transitions, summarized in Figure 10b. Initially, monoclinic distortions result in the formation of the O′3-Na_x_CrO_2_ phase through a first-order two-phase transition. This phase subsequently transforms into another monoclinic phase, O′3-E-Na_x_CrO_2_ (where ‘E’ denotes an expanded interlayer distance), before transitioning to the monoclinic P′3-Na_x_CrO_2_ phase. Despite the asymmetry between charge and discharge processes, the overall O3 ⇋ P′3 phase transitions are reversible. Furthermore, O3-NaCrO_2_ maintains good cycling performance when less than 0.5 Na ions are extracted.

When charged beyond 3.5 V, Jakobsen et al. [43] identified an oxidation peak at 3.67 V in the CV profile. If the charge does not exceed this peak (Figure 10c), the subsequent cycles closely replicate the initial cycle, demonstrating stability in the electrochemical response. Conversely, when the charge surpasses 3.67 V (Figure 10d), subsequent cycles deviate significantly from the initial one, exhibiting broader peaks indicative of increased solid-solution behavior. Despite these changes, Na intercalation remains reversible.

At voltages approaching 4.0 V, corresponding to the complete removal of Na ions, an additional oxidation peak emerges at approximately 3.85 V. As shown in Figure 10e, deep charging to this voltage induces irreversibility in Na_x_CrO_2_, as evidenced by the absence of reduction peaks during the discharge process. This suggests that the extraction of the final Na ions disrupts the structural integrity of the material, rendering the intercalation mechanism non-reversible.

Yabuuchi et al. [44] investigated the thermal stability of the NaCrO_2_ cathode material. Their results showed that NaCrO_2_ exhibited no phase transitions during heating in the temperature range of 27–527 °C, with only thermal expansion observed. The thermal expansion of NaCrO_2_ was significantly lower than that of LiNi_0.5_Mn_0.5_O_2_. Specifically, the linear thermal expansion coefficients for NaCrO_2_ were 0.90 × 10^−5^ °C^−1^ (a_hex_) and 1.48 × 10^−5^ °C^−1^ (c_hex_), compared to 1.68 × 10^−5^ °C^−1^ (a_hex_) and 2.21 × 10^−^^5^ °C^−1^ (c_hex_) for LiNi_0.5_Mn_0.5_O_2_. The thermal expansion of NaCrO_2_ was anisotropic, with the ratio of a_hex_/c_hex_ increasing as the temperature rose. Furthermore, oxygen loss in P3-NaCrO_2_ occurred at significantly higher temperatures (above 500 °C) compared to Ni-based cathode materials in LIBs, which experience oxygen loss between 240 and 260 °C.

In order to improve structural stability of NaCrO_2_ cathode materials, elemental doping has been an interesting strategy, for example, Ti [55], Sn [56], Ru [57], Nb [58], Mn [59], Al [60], and Ca [46,61]. Ma et al. [62] explored the effect of Sb substitution on the structural stability of Na_1-2x_Cr_1-x_Sb_x_O_2_ (x = 0, 0.05, 0.1, and 0.2) as a cathode material for SIBs. The Sb^5+^ ion, a semiconducting metal ion with an ionic radius (0.0600 nm) comparable to Cr (0.0615 nm), enhances structural stability due to its high valence state and the introduction of electron holes, which modify the local electronic environment. Their findings indicated that substituting a suitable amount of Sb, specifically in Na_0.90_Cr_0.95_Sb_0.05_O_2_ (NCS-0.05), significantly improved the material’s cycling and rate performance.

NCS-0.05 demonstrated exceptional cycling stability, retaining 70.9% of its initial capacity after 1000 cycles, in contrast to the 46.8% and 47.8% capacity retention observed for NaCrO_2_ and Na_0.8_Cr_0.9_Sb_0.1_O_2_ (NCS-0.1), respectively (Figure 11a). At a low current density of 0.1 C, NCS-0.05 exhibited a reversible capacity of approximately 120 mAh g^−1^. While the capacity decreased with increasing Sb substitution in the initial cycles, as the current density increased from 0.1 C to 32 C, the capacities of NaCrO_2_ (NC), NCS-0.05, and NCS-0.1 declined to 31.3, 96.1, and 69.3 mAh g^−1^, respectively. Upon reverting the current density to 0.1 C, their capacities recovered near their initial values. This comparison highlights that NCS-0.05 exhibited the best rate capability among the samples (Figure 11b). Based on the characterization results, the increase in the c/a ratio due to Sb substitution expands the Na^+^ diffusion channel, facilitating enhanced sodium diffusion. This structural adjustment is advantageous for improving the reversibility of Na^+^ intercalation and de-intercalation processes.

Ti doping has been reported for both P2- and O3-type Na_x_CrO_2_ cathode materials in SIBs [55,63,64]. Li et al. [64] investigated the electrochemical performance of O3-Na_1-x_Cr_1-x_Ti_x_O_2_ (with x = 0.03, 0.05, and 0.10). Although Ti doping reduces the initial capacity of the material, it significantly enhances the initial coulombic efficiency due to improved structural stabilization and the expansion of the NaO_2_ interlayer spacing, attributed to an increase in the c/a lattice ratio. Long-term cycling performance at a rate of 1 C revealed that as x increases, the initial specific discharge capacities of the samples decrease (103.7, 100.6, 96.7, and 88.8 mAh g^−1^ for x = 0, 0.03, 0.05, and 0.10, respectively). However, the corresponding capacity retentions improve substantially (50.0%, 71.6%, 80.1%, and 85.8% after 800 cycles, respectively, Figure 11c). This demonstrates that Ti doping greatly enhances the cycling performance of NaCrO_2_, further validating the improved structural stability with increasing Ti content. Additionally, the Ti-doped samples exhibit superior rate performance. When tested at a high rate of 30 C (3 A g^−1^), Na_0.95_Cr_0.9_5Ti_0.05_O_2_ delivers a reversible capacity of 67 mAh g^−1^, significantly outperforming the 42 mAh g^−1^ capacity of undoped NaCrO_2_ (Figure 11d). The improved stability of Ti-doped samples is attributed to the higher bond dissociation energy of the Ti-O bond (661.9 kJ mol^−1^) compared to the Cr-O bond (427 kJ mol^−1^) at 298 K. This stronger bond suppresses TMO_2_ layer gliding during cycling, delaying the phase transition from the hexagonal O3 structure to the monoclinic P3 structure, thereby enhancing the material’s structural integrity and electrochemical performance.

**Figure 11 micromachines-16-00137-f011:**
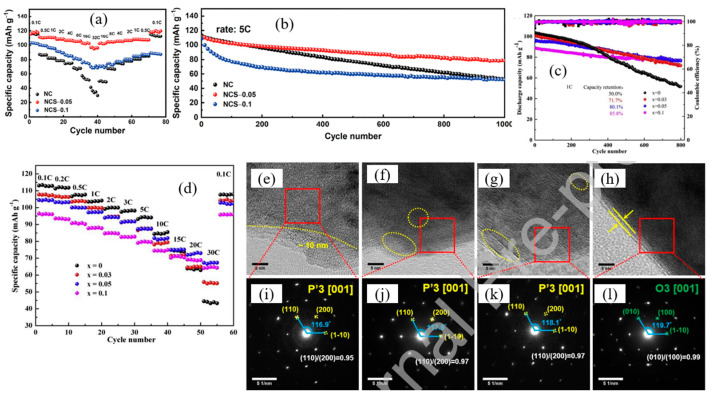
(**a**) Rate capability of NC, NCS-0.05, and NCS-0.1 at rates ranging from 0.1 C to 32 C, highlighting the superior performance of NCS-0.05 under high current densities, (**b**) cycling performance at 5 C for NC, NCS-0.05, and NCS-0.1, with NCS-0.05 exhibiting the highest capacity retention over prolonged cycles. Reprinted from Ref. [62] with permission from Elsevier, (**c**) cycling performance of Na_1-x_Cr_1-x_Ti_x_O_2_ at a rate of 1 C, indicating the enhanced capacity retention of Ti-doped samples over prolonged cycling, and (**d**) rate capability of Na_1−x_Cr_1−x_Ti_x_O_2_ samples, tested at rates ranging from 0.1 C to 30 C, highlighting the superior rate performance of Ti-doped materials compared to undoped NaCrO_2_. Reprinted from Ref. [64] with permission from Elsevier. Ex situ HR-TEM images of (**e**) NaCrO_2_, (**f**) Na_0.97_Cr_0.97_Ti_0.03_O_2_, (**g**) Na_0.93_CrCa_0.035_O_2_, and (**h**) Na_0.9_Ca_0.035_Cr_0.97_Ti_0.03_O_2_ after 1000 cycling at 10 C. SAED pattern of (**i**) NaCrO_2_, (**j**) Na_0.97_Cr_0.97_Ti_0.03_O_2_, and (**k**) Na_0.93_CrCa_0.035_O_2_ along the P′3 [001] zone axis, and (**l**) Na_0.9_Ca_0.035_Cr_0.97_Ti_0.03_O_2_ along the O3 [001] zone axis, respectively. Reprinted from Ref. [61] with permission from Elsevier.

Lee et al. [61] investigated the effects of Ca and Ti co-substitution in the NaCrO_2_ structure and found that it significantly improved cycling performance at a high current discharge capacity of 10 C. Based on ex situ TEM images (Figure 11e–h), structural damage and deformation were observed in pure NaCrO_2_, as well as in samples with single-element substitution (Ca or Ti). In contrast, the co-substituted sample exhibited a smooth surface, indicating minimal distortion in the layered structure, even at the particle edges, which are typically the most sensitive locations.

The structural damage in the other samples was attributed to large variations in unit cell volume and irreversible phase transitions during cycling. Selected area electron diffraction (SAED) analysis (Figure 11i–l) further confirmed the structural stability of Na_0.9_Ca_0.035_Cr_0.97_Ti_0.03_O_2_. The angle between the (010) and (1 –10) planes for this sample remained close to that of the O3-NaCrO_2_ structure (119.7° compared to 120°), demonstrating excellent structural retention over cycling. In contrast, the corresponding angles for cycled NaCrO_2_, Na_0.93_CrCa_0.035_O_2_, and Na_0.97_Cr_0.97_Ti_0.03_O_2_ were 116.9°, 117.9°, and 118.1°, respectively, values closer to the P3’-Na_0.4_CrO_2_ structure, indicating less structural stability.

Surface modification has been widely recognized as an effective approach to enhance the structural stability of NaCrO_2_ cathode materials. Techniques such as Cr_2_O_3_ [60,65] and carbon coatings [66,67,68] have been frequently employed in various studies. Ding et al. [69] investigated the effects of carbon coating on NaCrO_2_ cathode materials to improve their stability.

The charge and discharge curves for both uncoated (naked) and carbon-coated NaCrO_2_ electrodes at different current rates are shown in Figure 12a. While the voltage profiles of the carbon-coated and naked electrodes are similar, the carbon-coated electrode exhibits a lower voltage polarization between the charge and discharge profiles. At a current rate of 5 mA g^−1^, the carbon-coated NaCrO_2_ delivers higher charge and discharge capacities of 135 mAh g^−1^ and 116 mAh g^−1^, respectively, compared to 125 mAh g^−1^ and 106 mAh g^−1^ for the naked electrode. However, an increase in current rate leads to higher polarization and rapid capacity fading for both electrodes.

Figure 12b presents the cycling performance at a current rate of 5 mA g^−1^. Both the carbon-coated and naked electrodes initially show an increase in discharge capacity, reaching maximum values of 118 mAh g^−1^ and 110 mAh g^−1^ in the third and fourth cycles, respectively. The naked NaCrO_2_, however, exhibits faster capacity fading during cycling. After 40 cycles, the carbon-coated NaCrO_2_ retains a discharge capacity of 110 mAh g^−1^, which is notably higher than that of the uncoated electrode. This highlights the effectiveness of carbon coating in enhancing the cycling stability of NaCrO_2_ cathode materials.

Yu et al. [19] explored the effect of carbon coating on NaCrO_2_ cathodes and observed a significant enhancement in conductivity, increasing from 4.4 × 10^−3^ S cm^−1^ for the bare sample to 4.7 × 10^−1^ S cm^−1^ for a 14.6 wt% carbon-coated sample (Figure 12c). The initial charge–discharge profiles of bare NaCrO_2_ and carbon-coated NaCrO_2_ (C-NaCrO_2_, 3.4 wt%) were similar, except for the discharge capacities, which were 112 mAh g^−1^ and 121 mAh g^−1^, respectively. The carbon coating facilitated greater capacity delivery above 3.07 V during charging, attributed to a reversible phase transformation from an O3 to a P3 structure (Figure 12d).

Bare NaCrO_2_ exhibited significant capacity fading during cycling, while C-NaCrO_2_ electrodes demonstrated excellent stability, retaining over 96% of their capacity after 50 cycles for coatings ranging from 1.4 to 14.2 wt%. Among these, the 3.4 wt% C-NaCrO_2_ electrode displayed the best performance (Figure 12e).

The rate capability of C-NaCrO_2_ was particularly remarkable (Figure 12f). While bare NaCrO_2_ delivered only 3 mAh g^−1^ at a high rate of 50 C, the 3.4 wt% and 7.3 wt% C-NaCrO_2_ electrodes delivered capacities of 106 mAh g^−1^ at 50 C (5.5 A g^−1^), retaining 87% of the capacity at 0.5 C (55 mA g^−1^). The performance extended to an extremely high rate of 150 C (16.5 A g^−1^), with a 27 s discharge delivering 99 mAh g^−1^, demonstrating that the capacity was solely linked to Na^+^ insertion, as evidenced by the flat discharge profile (Figure 12g).

Table 2 compares the electrochemical performance of Cr-based cathode materials in SIBs to provide better clarification of their behavior and capabilities.

### 2.2. NaFeO_2_

Minor-metal-free LiFeO_2_ is unsuitable as a cathode material for LIBs due to its electrochemical inactivity. The oxidation of Fe^3+^ to Fe^4+^ is challenging because oxygen evolution precedes Fe^3+^ oxidation, unlike the Fe^2+^ to Fe^3+^ oxidation observed in LiFePO_4_ [71]. Additionally, the similar ionic radius of Fe^3+^ and Li^+^ promote cation mixing. An empirical rule suggests that layered AMO_2_ compounds are stabilized when the ionic radius ratio of the alkali metal (r_A_) to the transition metal (r_M_), r_A_/r_M_, exceeds 1.23. For LiFeO_2_ (r_Li_/r_Fe_ = 1.18), this criterion is unmet, leading to structural instability [72]. In contrast, NaFeO_2_ (r_Na_/r_Fe_ = 1.58) avoids such issues due to the larger Na^+^ ion. NaFeO_2_ exists as orthorhombic β-NaFeO_2_, which is electrochemically inactive, and rhombohedral α-NaFeO_2_, a cathode with O3-type layered structure per Delmas’ notation [73,74,75,76].

Okada and co-workers [73,77] were the first to report the reversible sodium cycling of α-NaFeO_2_, synthesized by solid-state reaction of equal molar ratio Fe_2_O_3_ and Na_2_O_2_ at 650 °C in air, in sodium-ion cells using a sodium metal counter electrode. Charging up to 3.6 V vs. Na yielded an initial charge capacity of 103 mAh g^−1^, corresponding to the formation of Na_0.58_FeO_2_, assuming no parasitic reactions. The cell discharged reversibly to 1.5 V vs. Na with an initial discharge capacity of 85 mAh g^−1^ and exhibited a flat operating voltage plateau at ~3.4 V. Sodium iron oxide has garnered interest as a cathode material for SIBs due to its high operating voltage (~3.3 V) based on the Fe^3+^/Fe^4+^ redox reaction and a capacity of ~80 mAh g^−1^. Iron’s abundance makes it appealing for SIBs; however, challenges arise at high potentials, including structural instability of the cathode and electrolyte decomposition. These issues cause rapid performance degradation due to irreversible structural transitions, particularly the formation of monoclinic Na_0.5_FeO_2_ during desodiation beyond x > 0.5 in Na_1-x_FeO_2_ [38,73,78,79].

Tetravalent iron (Fe^4+^) is rare but has been observed in certain perovskite oxides synthesized under high oxygen pressure. Mössbauer spectroscopy of α-NaFeO_2_ and its charged states reveals that it operates via the Fe^3+^/Fe^4+^ redox couple. The initial NaFeO_2_ exhibits high-spin Fe^3+^ characterized by an isomer shift (IS) of 0.36 mm s^−1^ and quadrupole splitting (QS) of 0.50 mm s^−1^, indicating a uniform Fe^3+^ valence state. Upon charging, an increase in QS suggests distortion of FeO_6_ octahedra. Signals for Fe^4+^ (IS = 0.05 mm s^−1^, QS = 0.55 mm s^−1^) were observed, with 18% of the iron identified as Fe^4+^ at 3.6 V, demonstrating the feasibility of sodium-ion deintercalation and Fe^3+^ oxidation (Figure 13a) [73,80]. Takeda et al. [81] corroborated these findings through Mössbauer analysis of NaFeO_2_, confirming Fe^4+^ formation during Na extraction.

The O3-NaFeO_2_ structure comprises Na and Fe octahedrally coordinated with oxygen, forming Na layers between O-Fe-O slabs (Figure 13b). During the initial charge to 4.4 V, the voltage profile exhibits plateaus at ~3.3 V and ~3.8 V, corresponding to Fe^3+^ oxidation and possible oxygen redox, respectively. Approximately 64% of Na-ion extraction (150 mAh g^−1^) occurs during the 3.3 V plateau. However, significant polarization (≥1.0 V) and low coulombic efficiency (~42%) were observed during discharge, with voltage plateaus shifting to 2.5 V and 2.3 V. Subsequent cycles show diminished capacity and increased polarization, with the Fe^3+^/Fe^4+^ redox reaction restricted to the first cycle (Figure 13c). This degradation is attributed to severe structural changes, including Fe migration, highlighting the challenges of using NaFeO_2_ as a cathode material.

Yabuuchi et al. [82] investigated the influence of cutoff voltage on the structural and electrochemical performance of the α-NaFeO_2_ cathode in sodium-ion cells. Their findings indicate that the reversibility of the electrode material is significantly affected by the cutoff voltage during charging (sodium extraction), as illustrated in Figure 14a. While a higher cutoff voltage increased the sodium extraction capacity, it simultaneously reduced the reversible capacity. Optimal performance, with minimal polarization, was achieved at a cutoff voltage of 3.4 V. Beyond this value, charging to 4.0 V or higher significantly increased polarization and reduced reversible capacity, rendering the material nearly inactive as an electrode at higher potentials. Approximately 70% of sodium can be extracted from O3-type NaFeO_2_ by oxidation up to 4.5 V, assuming no side reactions. However, this degree of sodium extraction compromises the electrode’s electrochemical activity and structural integrity (Figure 14a). The cyclability of the O3-type NaFeO_2_ at various cutoff voltages was further examined (Figure 14b,c). Cells charged to different voltages between 3.4 and 4.5 V, then discharged to 2.5 V at a current rate of 12.1 mA g^−1^, demonstrated varying degrees of performance. A cutoff voltage of 3.4 V maintained good capacity retention, with an initial reversible capacity of 80 mAh g^−1^ and approximately 75% capacity retention after 30 cycles (Figure 14c). Increasing the cutoff voltage to 3.5 V enhanced the first-cycle reversible capacity to 100 mAh g^−1^ but slightly reduced long-term capacity retention compared to the 3.4 V cycle. Further increases in the cutoff voltage to 4.0 V or higher led to significant capacity fade and poor overall cyclability.

Ex situ XRD analysis was conducted to elucidate the mechanism underlying the poor reversibility observed at higher cutoff voltages (Figure 14d). At a cutoff voltage of 3.4 V, sodium ions were reversibly extracted in a topotactic manner. An increase in the intensity of the (003)_hex_ diffraction line relative to the (104)_hex_ line after cycling between 3.4 and 2.5 V indicates the formation of sodium-ion vacancies at 3b octahedral sites. In O3-type sodium-layered compounds, phase transitions from O3 to P3 are commonly observed during sodium extraction when x > 0.1 in Na_1-x_MeO_2_. These transitions, caused by sodium/vacancy ordering at 3b sites, lead to stepwise voltage changes during galvanostatic oxidation/reduction. However, in Na_1-x_FeO_2_, the cell voltage increases monotonically with sodium extraction.

Irreversible structural changes occur when the charging voltage exceeds 4.0 V (x > 0.5 in Na_1-x_FeO_2_). At 4.5 V, the irreversible capacity increases significantly, likely due to the formation of additional vacancies at 3b sites. This is accompanied by a reduction in the intensity of the (003)_hex_ diffraction line relative to the (101)_hex_ and (104)_hex_ lines. Such changes suggest possible iron migration from 3a to 6c (face-shared tetrahedral to octahedral sites) or to 3b sites, assuming no oxygen release from the lattice. This migration of iron ions may block sodium conduction pathways, contributing to the observed decline in electrode performance.

The chemical instability of Fe^4+^ in NaFeO_2_ cathode materials further exacerbates performance issues. Lee et al. [83] demonstrated that the electrochemical performance of NaFeO_2_ significantly deteriorates when stored in a charged state. In their study, a coin cell charged to 3.6 V, stored for 24 h, and then discharged to 2.0 V exhibited a much lower specific capacity (62.7 mAh g^−1^) and average cell voltage compared to a cell discharged immediately after charging (80 mAh g^−1^). This decline is attributed to the chemical instability of Fe^4+^, which spontaneously reduces Fe^3+^ in the charged state, accompanied by electrolyte decomposition.

Given that batteries are often stored in fully or partially charged states, this instability leads to significant self-discharge and rapid performance degradation. Addressing the chemical instability of Fe^4+^ is thus critical to unlocking the potential of Fe^3+^/Fe^4+^-based redox cathodes.

The thermal stability of NaFeO_2_ is promising compared to cathodes used in LIBs. When NaFeO_2_ was heated to 450 °C, no exothermic reactions were observed. In contrast, exothermic reactions were reported for Li_0.5_CoO_2_—a widely used cathode material in portable devices—at temperatures of 190 °C and 395 °C. However, it is noteworthy that heat generation occurred in the case of charged NaFeO_2_ (Na_0.58_FeO_2_) above 350 °C, as observed in differential scanning calorimetry (DSC) measurements, according to reaction 3 [73]:Na_0.58_FeO_2_ → 0.58 NaFeO_2_ + 0.21 Fe_2_O_3_ + 0.105 O_2_
(3)

Elemental doping has proven to be an effective strategy for enhancing the performance of NaFeO_2_ cathode materials [84]. Xu et al. [85] incorporated Ru as a doping element into NaFeO_2_, demonstrating that Ru-doped NaFeO_2_ (Na_4_FeRuO_6_) delivered a high reversible capacity of 120 mA h g^−1^ and achieved approximately 80% capacity retention after 100 cycles when charged up to 4 V. Furthermore, Ru doping significantly reduced potential polarization in NaFeO_2_.

The typical charge–discharge cycles of Na_4_FeRuO_6_ and NaFeO_2_ are presented in Figure 15a,b. NaFeO_2_ delivered about 85 mAh g^−1^ reversible capacity at 0.2 C during the initial charge–discharge cycle, accompanied by a low coulombic efficiency of 81.7%. In contrast, Na_4_FeRuO_6_ exhibited a reversible capacity of 120 mA h g^−1^ within a voltage range of 2–4 V, with a much higher coulombic efficiency of 96.7%, indicating improved reversibility and reduced polarization.

Na_4_FeRuO_6_ demonstrated superior rate capability, delivering capacities of 120 mAh g^−1^ at 0.2 C, 96 mAh g^−1^ at 0.5 C, 88 mAh g^−1^ at 1 C, 75 mAh g^−1^ at 2 C, and 55 mAh g^−1^ at 5 C. In contrast, NaFeO_2_ showed poor rate performance, with negligible capacity at 2 C (Figure 15c,d). The improved performance of Ru-doped NaFeO_2_ is attributed to the suppression of Fe-ion migration, which reduces internal strain and mitigates particle cracking associated with the structural distortions caused by Fe-ion migration.

Co-doping in NaFeO_2_ has been shown to significantly enhance its electrochemical performance, as demonstrated by Yoshida et al. [86]. The study revealed that although NaFe0.5Co_0.5_O_2_ is considered an intermediate solid solution between NaFeO_2_ and NaCoO_2_, it delivers the highest reversible capacity of approximately 160 mAh g^−1^ among the three materials tested. Notably, the average operating voltage of NaFe_0.5_Co_0.5_O_2_ (3.14 V) in Na cells exceeds that of NaCoO_2_ (2.96 V) (Figure 15e).

Additionally, both NaFe_0.5_Co_0.5_O_2_ and NaCoO_2_ exhibit relatively good capacity retention over cycling, as illustrated in Figure 15f. A remarkable finding is the superior rate capability of NaFe_0.5_Co_0.5_O_2_, which significantly outperforms NaCoO_2_. Even at a high rate of 30 C (7.23 A g^−1^), micrometer-sized NaFe_0.5_Co_0.5_O_2_ achieves a discharge capacity exceeding 100 mAh g^−1^, highlighting its excellent fast-charging and high-rate performance.

Shevchenko et al. [87] investigated the effect of Mn and Ni substitution on the performance of NaFeO_2_-based cathodes, synthesizing NaFe_1/3_Mn_1/3_Ni_1/3_O_2_ to reduce Fe ion migration into the Na layers. The electrochemical performance of this cathode material was evaluated by charging to 4.0 V and 4.2 V and discharging to 1.9 V. At a C/10 rate, the material delivered specific capacities of approximately 130 mAh g^−1^ and 160 mAh g^−1^, respectively. However, charging up to 4.2 V introduced a plateau around 4.1 V, which led to capacity degradation over cycling due to structural instability.

### 2.3. NaCuO_2_

Copper is abundant, relatively inexpensive, and among the less toxic transition metals. Despite its favorable properties, there have been no reported electrochemical studies on LiCuO_2_. Efforts to synthesize LiCuO_2_ via solid-state calcination have been unsuccessful, as the reaction results in the formation of Li_2_CuO_2_ and CuO, as shown below:2 CuCO_3_ + Li_2_CO_3_ → Li_2_CuO_2_ + CuO + 3 CO_2_
(4)

In contrast, NaCuO_2_ has been successfully synthesized using the solid-state calcination of Cuo and Na_2_O_2_ at 450 °C in oxygen environment. This behavior parallels that of nickelates, where NaNiO_2_ can be readily prepared via solid-state calcination, while stoichiometric LiNiO_2_ is notably challenging to obtain. Structural studies reveal that NaCuO_2_ adopts a monoclinic structure with the same space group and elemental arrangement as LiCuO_2_, as illustrated in Figure 16a. This structural similarity indicates that NaCuO_2_ may hold promise as a positive electrode material for sodium-ion batteries.

Figure 16b presents the discharge–charge profiles of a Na/NaCuO_2_ cell from the first to the fifth cycle within the voltage range of 0.75 to 3.0 V [88].

Theoretically, sodium-ion insertion and extraction in NaCuO_2_ correspond to the reduction of Cu^3+^ to Cu^2+^ during discharge and the oxidation of Cu^3+^ to Cu^4+^ during charge, respectively.NaCu^3+^O_2_ + x Na^+^ + x e^−^ → Na_1+x_Cu^2+^O_2_
(5)NaCu^3+^O_2_ → Na_1-x_Cu^4+^O_2_ + x Na^+^ + x e^−^
(6)

The subsequent charge capacity of NaCuO_2_ reached 142 mAh g^−1^, nearly equivalent to its initial discharge capacity. Inductively Coupled Plasma Atomic Emission Spectroscopy (ICP-AES) analysis indicated that sodium-ion insertion and extraction during the first discharge and charge processes corresponded to 0.8 mol and 0.5 mol, respectively. In the case of LiCuO_2_, lithium-ion insertion occurs at approximately 2.0 V (vs. Li), leading to the formation of Li_1.7_CuO_2_.

Despite stable discharge–charge plateaus over the first five cycles, a noticeable decline in discharge capacity was observed with continued cycling. This significant capacity fading suggests that structural changes occur in NaCuO_2_ during repeated electrochemical cycling, ultimately impacting its performance. Ono et al. [89] investigated the poor cycling performance of NaCuO_2_ in the voltage range of 1–3 V and identified the underlying mechanism. Sodium-ion insertion into NaCuO_2_ during the first discharge results in an irreversible conversion of NaCuO_2_ to an amorphous phase, Na_2_CuO_2_. Subsequent charge and discharge reactions proceed between these amorphous phases. However, the discharge product, Na_2_CuO_2_ (more accurately Na_1.6_CuO_2_), is unstable and undergoes decomposition, forming copper oxide (CuO) and sodium oxide (Na_2_O):NaCuO_2_ + Na^+^ + e^−^ → Na_2_CuO_2_
(7)Na_2_CuO_2_ → Na_2_O + CuO (8)
subsequently, CuO reacts with sodium ions, producing Cu_2_O and additional Na_2_O:(9)$$\mathrm{CuO}+{\mathrm{Na}}^{+}+e^{-}\to \frac{1}{2}{\mathrm{Cu}}_{2}O+\frac{1}{2}{\mathrm{Na}}_{2}O$$

These irreversible reactions contribute to the significant capacity fading observed in NaCuO_2_ during repeated electrochemical cycling.

### 2.4. NaNiO_2_

The structure of NaNiO_2_ was first identified in 1954 using film methods with a precession camera and later refined through Rietveld analysis with X-ray and neutron scattering in 1977. Subsequent studies in 2000 and 2005 employed neutron and X-ray diffraction, respectively, to further understand its structure [90,91,92,93]. NaNiO_2_ exhibits two polymorphs: a low-temperature monoclinic O3 layered structure and a high-temperature rhombohedral phase. The low-temperature phase features NiO_6_ octahedra that are edge-sharing and elongated due to the Jahn–Teller distortion of the Ni^3+^ ions. Sodium ions occupy interlayer positions, forming distorted octahedral coordination with oxygen atoms, and this phase is isostructural with NaMnO_2_. At approximately 450 K, thermal motion overcomes the static Jahn–Teller distortion, causing a phase transition to the rhombohedral form, which is isostructural with α-NaFeO_2_ and LiNiO_2_. The monoclinic NaNiO_2_ polymorph was synthesized by solid-state reaction of excess amounts of Na_2_O and NiO at 650 °C under O_2_ flow and tested as a cathode material, exhibiting a stepwise charge–discharge profile with an initial open-circuit voltage of 2.47 V. Vassilaras et al. [94] demonstrated, in contrast to previous studies (which reported during the first electrochemical cycle, only 0.2 sodium ions could be extracted from NaNiO_2_ [94,95]), a charge capacity of 199 mAh g^−1^ (x = 0.85 in Na_x_NiO_2_) and a discharge capacity of 147 mAh g^−1^ (x = 0.62 in Na_x_NiO_2_) within the voltage range of 2.0–4.5 V. Galvanostatic charge and discharge measurements of NaNiO_2_ at C/10 (1 C = 235 mAh g^−1^) were conducted over multiple cycles in two different voltage ranges: 1.25–3.75 V and 2.0–4.5 V. The results are illustrated in Figure 17a,b.

The charge and discharge curves of NaNiO_2_ exhibit the expected step-like features, indicating distinct phase transitions during sodium extraction and insertion. Except for the first charge cycle, the similarity in the plateaus suggests consistent phase transitions throughout subsequent cycling. In the voltage range of 1.25–3.75 V (Figure 17a), the charge and discharge capacities for the first, second, tenth, and twentieth cycles are 147/123, 121/119, 119/118, and 117/116 mAh g^−1^, respectively, with corresponding coulombic efficiencies of 83.8%, 98.4%, and 99.2% for the first, second, and subsequent cycles [94].

When the voltage range is extended to 2.0–4.5 V (Figure 17b), the charge and discharge capacities for the first, second, tenth, and twentieth cycles are 199/147, 136/128, 127/107, and 113/98 mAh g^−1^, respectively. The coulombic efficiencies under this range are 73.7%, 93.7%, 84.6%, and 86.3% for the first, second, tenth, and twentieth cycles, respectively. A significant decline in capacity and coulombic efficiency is observed at higher voltages, indicative of side reactions and structural instability.

The first charge curve deviates from subsequent charge cycles, particularly below 3.0 V. However, above 3.0 V, the charge profiles align with those of later cycles, suggesting that excess capacity in the first cycle is associated with reactions at lower voltages. Cyclic voltammetry analysis confirms the capacity decline and differences between charge and discharge profiles over extended cycling at higher voltages (Figure 18a). The CV curves reveal four reversible oxidation and reduction peaks at 2.7/2.3, 3.1/2.9, 3.4/3.2, and 3.5/3.4 V. Beyond the first cycle, subsequent scans show consistent reversibility within the 2.0–4.0 V range.

The oxidation peaks between 2.5 and 3.0 V during the first cycle differ from those in later cycles, corroborating the galvanostatic cycling data. Above 4.0 V, oxidation processes appear partially reversible, as no corresponding reduction peaks are observed near 4.0 V. These observations suggest that side reactions at higher voltages contribute to poor cycling performance. Both galvanostatic and CV results highlight distinct reactions occurring in the 2.5–3.0 V range during the initial charge cycle, which are absent in subsequent cycles.

To further understand structural changes after the first cycle, XRD analysis was performed, and the diffraction patterns of three samples are compared in Figure 18b.

The bottom XRD pattern (black) corresponds to the pristine NaNiO_2_ sample. The middle pattern (red) represents the XRD of the as-prepared electrode, which consists of NaNiO_2_ mixed with PTFE and carbon black. In addition to the pristine NaNiO_2_ phase, a secondary phase, identified as Na_0.91_NiO_2_, is observed. This indicates a small loss of sodium during electrode preparation, likely caused by air exposure or interactions with the binder and carbon black. The top pattern (blue) shows the XRD of the electrode after discharge following cycling between 2.0 and 3.75 V. Only the Na_0.91_NiO_2_ phase is present, with no evidence of the original NaNiO_2_ phase, suggesting that the discharge process does not fully restore the sodiated NaNiO_2_ phase.

Further analysis was conducted at higher voltages to investigate the observed capacity reduction. XRD patterns recorded after charging NaNiO_2_ to 3.75 V and 4.5 V are displayed in Figure 18c. While the majority of the patterns remain similar, the highly charged sample exhibits two additional peaks at approximately 23.7° and 35.0°. These peaks indicate the formation of a new phase, although its structure remains unidentified [94].

Han et al. [96] revisited the phase transitions of NaNiO_2_ during charge–discharge cycling within a voltage range of 1.5–4.0 V at a rate of C/10 (Figure 19a). During the first charge, four distinct plateaus were observed, reflecting multiple phase transitions. These transitions are attributed to gliding of the oxide layers, variations in structural distortion due to sodium deintercalation, and rearrangement of sodium ions within the interlayer space. The OCVs associated with these plateaus are 2.59 V, 3.04 V, 3.38 V, and 3.53 V, respectively.

During discharge, four stepwise plateaus similar to those during charge were observed, indicating that the phase transitions are highly reversible throughout the electrochemical process. However, the first discharge capacity is notably lower than the first charge capacity, a well-documented phenomenon attributed to incomplete re-sodiation during the first discharge. This results in significant capacity loss.

Additionally, the lengths of the plateaus during charge and discharge are approximately identical, except for the first plateau in both processes. A pronounced voltage drop below 2.3 V suggests that a highly unfavorable phase transition is necessary to achieve the fully sodiated phase.

Real-time in situ XRD measurements of a NaNiO_2_ powder electrode were performed within the 2θ ranges of 15–18° and 31–38°, as shown in Figure 19b. The initial scan clearly identifies the pristine material as pure NaNiO_2_. The initial scan identifies the pristine material as pure NaNiO_2_. However, during the charging process, the fully sodiated phase diminishes rapidly, and partially desodiated phases emerge, evidenced by a shift of the (001) reflection toward lower 2θ angles. This shift corresponds to an increase in the c-spacing as sodium ions are deintercalated. The removal of positively charged sodium ions enhances electrostatic repulsion between the oxide layers, leading to structural expansion. Concurrently, the (−201) reflection shifts toward a smaller d-spacing, indicating contraction along the a- and b-axes. This shrinkage is attributed to the increasing concentration of smaller Ni^4+^ ions during oxidation.

As the charge progresses, new biphasic regions appear. Between sodium contents of ~1.0 and 0.74 per formula unit, O′3, P′3, and O’’’’3 phases emerge. The transition between these phases is accompanied by increased distortion, evidenced by changes in lattice parameters. Upon reaching a sodium concentration of 0.55 at 3.26 V, most O′3 and P′3 phases disappear, and the P″3 phase dominates. The a/b ratio increases from 1.73 to 1.76, reflecting enhanced structural distortion, though the β angle remains stable at 121.5°, preserving hexagonal symmetry.

Further oxidation at 3.38 V (sodium content ~0.55 to 0.42) reveals a new plateau corresponding to the coexistence of P″3, P‴3, and O″3 phases. During this stage, the P″3 phase transitions into a more distorted P3-like structure, before the formation of the O″3 phase. The newly identified P‴3 phase exhibits greater distortion than P″3, though it does not conform to the O″3 structural model. As the P‴3 phase transitions to O″3, the a/b ratio remains relatively stable, but the β angle shifts significantly from 119.75° to 106.0°, marking a transition from hexagonal to monoclinic symmetry. This phase evolution, driven by increased interlayer repulsion due to extensive sodium removal, exemplifies the transition from P-type to O-type phases.

A small biphasic region is observed between O″3 and O‴3 when the sodium content decreases from 0.42 to 0.40. This quickly transitions to a solid solution of O‴3. By 3.74 V in the relaxed in situ XRD scans, peaks corresponding to the O′3, P′3, P″3, and O″3 phases are nearly absent, leaving only the O‴3 phase detectable at sodium concentrations as low as 0.37. The lattice parameters ‘a’ and ‘c’ represent the diagonal and edge lengths of the MO_6_ octahedron, respectively, and relate to the hexagonal lattice parameters as follows:a (monoclinic) = √3 × a (hexagonal)b (monoclinic) = a (hexagonal)       

Pristine NaNiO_2_ exhibits an a/b ratio of 1.88 and a β angle of 110.45°, indicative of a significant Jahn–Teller distortion caused by the low-spin t^6^_2g_ e^1^_g_ electronic configuration of Ni^3+^ [92]. This distorted monoclinic phase is stable at room temperature, while the ideal rhombohedral phase stabilizes only above 450 K, where thermal energy overcomes the Jahn–Teller effect.

During discharge, the phase transitions and voltage steps mirror those observed during charging. However, as the discharge voltage approaches 2.3 V, polarization increases, indicating that sodium re-intercalation into the host structure becomes kinetically unfavorable near full sodiation. Similar observations were made by Wang et al. [97], who reported irreversible structural changes in NaNiO_2_ during charge–discharge cycling in the 2–4.5 V range. Two-dimensional phase mapping revealed that these structural changes significantly hinder the electrochemical reversibility of the material.

Notably, a steep voltage drop below 2.3 V and the emergence of an O’’’’3 phase (marked with an asterisk in Figure 19b) during discharge highlight the difficulty in achieving full sodiation. This behavior underscores the structural and kinetic limitations of NaNiO_2_ during prolonged cycling.

Kim et al. [98] investigated the thermal stability of Na_0.91_NiO_2_ and Na_0.5_NiO_2_ in the presence of an electrolyte within the temperature range of 300–600 °C, as shown in Figure 20a–d. Two phase transitions were observed in both samples during heating: the first transition occurred between 247 and 352 °C, and the second between 528 and 600 °C (Figure 20a,c). Both samples exhibited the formation of a Ni metal phase and a structural transition from a layered structure to rock salt (Figure 20b,d).

However, a key difference was identified between the two samples. The onset temperature for phase decomposition was lower for the desodiated sample (Na_0.5_NiO_2_), indicating that increased Na extraction from the structure reduces the decomposition temperature.

A substantial number of studies have explored the use of Ni as a primary transition metal in SIB cathode materials, often combined with reinforcing elements such as Ti, Mn, and Co. Various compositions have been successfully synthesized and reported, including: NaFe_0.2_Ni_0.4_Ti_0.4_O_2_, NaFe_0.4_Ni_0.3_Ti_0.3_O_2_ [99], NaNi_0.5_Mn_0.5_O_2_ [100], NaNi_0.5_Mn_0.3_Ti_0.2_O_2_ [101], NaNi_0.8_Co_0.15_Al_0.05_O_2_ [102], NaNi_0.815_Co_0.15_Al_0.035_O_2_ [103], NaNi_x_Fe_y_Mn_z_O_2_ [104], NaNi_x_Mn_1-x_O_2_ [105], NaFe_0.5_Ni_0.5_O_2_ [106,107], Na_0.8_[Ni_0.3_Co_0.2_Ti_0.5_]O_2_ [108], NaNi_0.5_Ti_0.5_O_2_ [107,109], Na(Ni_2/3_Sb_1/3_)O_2_ [110], NaNi_0.9_Ti_0.1_O_2_ [111], and Na_x_Ni_0.6_Co_0.4_O_2_ [112] have been successfully reported as cathode materials in SIBs.

NaNi_0.5_Mn_0.5_O_2_ has been widely studied as a promising cathode material for sodium-ion batteries (SIBs) [113,114,115,116,117]. However, the O3-NaNi_0.5_Mn_0.5_O_2_ phase undergoes multiple complex phase transitions (O3-O3′-P3-P3′) and exhibits multiple voltage platforms during the charge and discharge processes. These transitions result in significant unit cell volume changes, structural collapse, and sluggish kinetic performance [100].

Ti substitution for Mn has been shown to increase interslab distances and suppress phase transformations in the high-voltage region [101,118]. Yuan et al. [117] reported that La doping in the structure of NaNi_0.5_Mn_0.5_O_2_ improves stability and rate capability by reducing particle size, stabilizing the crystal structure, and enhancing Na-ion diffusion. This enhancement is attributed to the formation of strong La-O bonds, which expand the Na interlayer spacing.

Liang et al. [114] developed a core–shell structure to improve the cycling and rate performance of NaNi_0.5_Mn_0.5_O_2_. In this design, a highly stable P2-Na_2/3_MnO_2_ coating was deposited on the high-capacity NaNi_0.5_Mn_0.5_O_2_ shell. The reinforced structure demonstrated significant performance improvements, retaining 85.3% of its capacity after 150 cycles at 1C (125 mAh g^−1^), compared to 74.8% for the bare sample. Additionally, the core–shell structure exhibited excellent high-rate capability, delivering 103 mAh g^−1^ at 15C, which is notably higher than the 82 mAh g^−1^ achieved by the uncoated sample.

Meghnani et al. [119] developed a Na_3_PO_4_ coating on NaNi_0.815_Co_0.15_Al_0.035_O_2_ particles to enhance air stability and mitigate the formation of insulating compounds such as NaOH and Na_2_CO_3_. This modification improved the material’s viability as a cathode for SIBs. The Na_3_PO_4_ coating contributed to a highly stable structure, improved sodium intercalation/deintercalation ability, and facilitated reversible phase transformations during the discharge process. These factors collectively resulted in promising electrochemical performance of Na_3_PO_4_@NaNi_0.815_Co_0.15_Al_0.035_O_2_ over extended cycling, as demonstrated in Figure 21a,b.

Li et al. [120] synthesized O3-type Ni-rich NaNi_2/3_Mn_1/6_Fe_1/6_O_2_ (NNMF) at various calcination temperatures (800 °C, 830 °C, 850 °C, and 880 °C) as a promising cathode material for SIBs. The first-cycle charge–discharge curves of NNMF samples, tested within a voltage range of 1.5–4.2 V at 0.2 C (1 C = 160 mA g^−1^), are shown in Figure 21c. The first discharge-specific capacities for NNMF-800, NNMF-830, NNMF-850, and NNMF-880 were 113, 117, 226, and 58 mAh g^−1^, respectively.

The cycling performance after 100 cycles, as shown in Figure 21d, revealed significant differences among the samples. The NNMF-850 electrode exhibited a gradual decrease in discharge capacity from 226 mAh g^−1^ to 158 mAh g^−1^, corresponding to a capacity retention rate of ~70%, demonstrating excellent cycling stability. In contrast, NNMF-800, NNMF-830, and NNMF-880 showed capacity retention rates of 40%, 46%, and 41.7%, respectively, indicating poor cycling stability.

Figure 21e illustrates the discharge performance of the electrodes at various current densities. The NNMF-850 electrode displayed superior rate performance compared to the other samples. For NNMF-850, the discharge capacities at 0.2 C, 0.5 C, 1 C, and 2 C were 225, 193, 168, and 145 mAh g^−1^, respectively, and the capacity recovered to 193 mAh g^−1^ upon returning to 0.2 C, demonstrating excellent reversibility. On the other hand, the discharge capacities of NNMF-800, NNMF-830, and NNMF-880 decreased more significantly as the current density increased, indicating inferior rate capabilities.

The superior performance of NNMF-850 is attributed to the optimization of its crystallinity and morphology, which enhances its structural stability and Na^+^ diffusion, ensuring excellent cycling stability and rate capability.

### 2.5. NaCoO_2_

Investigations into Na_x_CoO_2_ (0.5 ≤ x ≤ 1) have identified four distinct phases: O3 (0.9 ≤ x ≤ 1, α phase), O′3 (x = 0.75, α′ phase), P3 (0.55 ≤ x ≤ 0.6, β phase), and P2 (0.64 ≤ x ≤ 0.74, γ phase). These phases depend on non-stoichiometry and synthesis conditions (The most common synthesis method is solid-state reaction of (non)stoichiometric Co_3_O_4_ and Na_2_CO_3_ at different temperatures, based on phase type). Notably, the P2 phase demonstrates superior reversibility as a cathode material and enhanced thermoelectric power [121,122,123,124].

Among these, the O3 phase demonstrates superior energy density at voltages below 3.5 V [125,126,127]. However, O3 materials are strongly hygroscopic, which poses challenges for processing in air.

P2 compounds, on the other hand, have garnered significant attention due to their higher sodium mobility compared to O3 compounds, as demonstrated in computational and experimental studies. The P2 structure forms in the stoichiometry Na_x_MO_2_ when 0.6 < x < 0.8, while the O3 phase is more stable at full sodiation (x = 1.0). Consequently, synthesizing P2 compounds requires a higher average transition metal oxidation state (M^3.2+^/M^3.4+^) than for O3. As a result, only three single-metal P2 compounds (M = V, Mn, and Co) have been synthesized to date, while others are obtained in binary or ternary systems incorporating high-valent ions. However, these high-valence elements often contribute minimally to the electrochemical performance, either being inactive or adding only to the low-voltage capacity [37]. The charge–discharge curves of O3, P3, and P2 sodium cobalt oxide are shown in Figure 22a. Despite differences in the stacking sequences of CoO_2_ layers, Na_x_CoO_2_ phases exhibit multiple voltage plateaus. Large reversible capacities are achieved for all three phases: 140 mAh g^−1^ for the O3 phase, 130 mAh g^−1^ for the P3 phase, and 120 mAh g^−1^ for the P2 phase.

Layered NaCoO_2_ stands out due to its excellent structural stability and superior cyclability compared to LiCoO_2_. The sodium-ion diffusion coefficient of Na_x_CoO_2_ (0.5–1.5 × 10^−10^ cm^2^ s^−1^) is comparable to that of lithium-ion diffusion in LiCoO_2_ (1 × 10^−11^ cm^2^ s^−1^) [128]. In LiCoO_2_, excessive lithium-ion removal can lead to a structural transformation from hexagonal to tetragonal, causing capacity degradation. In contrast, NaCoO_2_ benefits from sodium’s larger ionic radius, preventing occupation of tetrahedral sites, which avoids structural deformation and lowers the potential barrier for ion migration.

Among the various phases of Na_x_CoO_2_ (0.5 ≤ x ≤ 1), the P2 phase (0.6 ≤ x ≤ 0.74) exhibits superior reversibility as a cathode material due to its large trigonal prismatic sites that enhance sodium-ion intercalation and improve cycle life. However, its complex phase transitions—caused by the rotation of MO_6_ octahedra and M-O bond breaking—can also enhance its cyclic performance, making the P2 phase a focus of recent research [129].

**Figure 22 micromachines-16-00137-f022:**
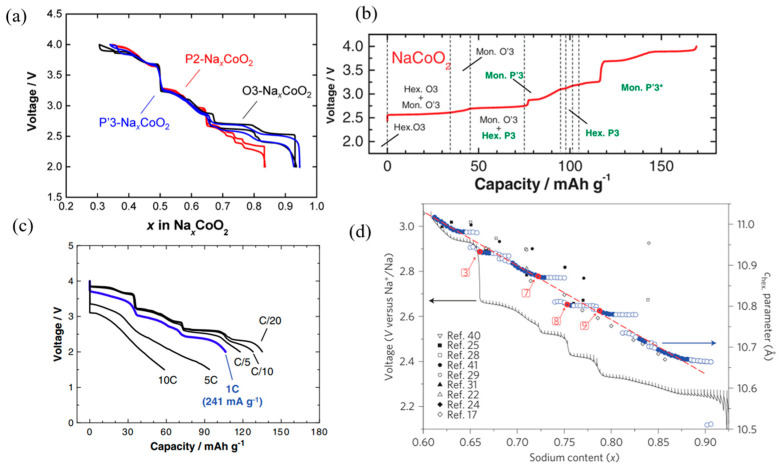
(**a**) Comparison of typical charge–discharge curves of Na cells with layered polymorphs: O3-, P′3-, and P2-type Na_x_CoO_2_. The curves show similar profiles across the three polymorphs, except in the region with high sodium content (x > 0.7), where differences become apparent. Reprinted from Ref. [130] with permission from the American Chemical Society, (**b**) galvanostatic charge curves and structural evolution of Na_1-x_CoO_2_, where detailed structural changes have not been reported in the monoclinic P′3 phase above 3.5 V. Reprinted from Ref. [38] with permission from Wiley, (**c**) rate capability of NaCoO_2_, indicating high capacity at low rates but diminished performance at higher current densities due to phase boundary movements and Na^+^/vacancy ordering. Reprinted from Ref. [86] with permission from Elsevier, and (**d**) evolution of the c_hex_ parameter as a function of sodium content in P2-Na_x_CoO_2_, showing its correlation with the electrochemical potential (red curve) and structural parameters in both single-phase (filled blue circles) and biphasic domains (open circles). In single-phase regions, the c_hex_ parameter shows an approximately linear evolution with sodium content (red dashed line), though deviations occur near phase composition boundaries, especially close to biphasic domains. Key single-phase compositions, corresponding to specific electrochemical transitions (drops no. 3, 7–9), are highlighted as red circles. Reprinted from Ref. [127] with permission from Nature.

Yoshida et al. [86] reported that O3-NaCoO_2_ delivers a discharge capacity of over 140 mAh g^−1^ within a voltage range of 2.5–4.0 V, featuring a stepwise voltage profile. This profile is attributed to multiple reversible phase transitions involving CoO_2_ slab gliding and in-plane vacancy-Na ordering (Figure 22b). The capacity loss during cycling in O3-NaCoO_2_ is likely due to the degradation of the electrolyte solution, exacerbated by the non-passivated surface of metallic sodium used as the counter electrode. Furthermore, the voltage profile of Na_x_CoO_2_ suggests a limited rate capability, attributed to phase boundary movements in two-phase regions and Na/vacancy ordering coupled with charge ordering (Figure 22c).

Unlike LiCoO_2_, the P2-type Na_x_CoO_2_ exhibits a highly intricate electrochemical charge–discharge profile characterized by several voltage plateaus. These plateaus are attributed to Na^+^/vacancy ordering, which arises from in-plane patterning between Na^+^ and vacancy sites, a phenomenon absent in the layered structure cathodes of LIBs. Na^+^/vacancy ordering results from energy minimization involving three types of cation interactions: Na^+^-Na^+^, Na^+^-Co^3+/4+^, and Co^3+/4+^-Co^3+/4+^, occurring below a critical temperature. This ordering is highly sensitive to the sodium content in the host material and is accompanied by a first-order phase transition at specific sodium concentrations. Such transitions introduce additional activation energy barriers for Na^+^ accommodation. Berthelot et al. [127] investigated the structural changes in P2-Na_x_CoO_2_ cathode materials and observed that the c_hex_ parameter decreases during discharge. This decrease is attributed to enhanced structural cohesiveness upon sodium intercalation. Simultaneously, a slight increase in the a_hex_ parameter-representing the M-M (metal-metal) distance within the slabs-is linked to the reduction of M^4+^ ions (Figure 22d). It can be observed that the voltage plateaus in the electrochemical curve correspond to biphasic transitions, during which the c_hex_ parameter remains constant for each phase. Conversely, in single-phase regions, the c_hex_ parameter evolves continuously with changes in sodium composition.

Similar to other sodium transition metal oxides, the doping of electrochemically (in)active elements has been explored as an effective strategy to enhance the performance of sodium cobalt oxide cathode materials. This approach aims to stabilize the structure, improve cycling stability, and optimize the electrochemical properties by mitigating issues such as phase transitions and structural degradation during sodium intercalation and deintercalation [37,131,132,133]. Kang et al. [134] investigated the effects of Ti doping in P2-type Na_0.67_Co_1-x_Ti_x_O_2_ (x < 0.2) cathodes under an extended potential range exceeding 4.4 V (Figure 23a–c). Their findings (Figure 23d) demonstrated that Na_0.67_Co_0.90_Ti_0.10_O_2_ exhibited excellent capacity retention of 115 mAh g^−1^ even after 100 cycles, while Na_0.67_CoO_2_ showed negligible capacity retention (<10 mAh g^−1^) under a 4.5 V cutoff voltage. Additionally, Na_0.67_Co_0.90_Ti_0.10_O_2_ displayed exceptional rate capabilities (Figure 23e), achieving 108 mAh g^−1^ at a current density of 1000 mA g^−1^ (7.4 C). These results suggest that the enhanced rate capability is primarily due to reduced kinetic resistance from Ti substitution, rather than the elimination of active sites for Na ions at high potential conditions above 4.4 V.

Ca doping has been identified as an effective way to enhance the cycling stability of P2-type Na_x_CoO_2_ cathode materials [135,136]. Han et al. [136] explored the cycling and rate performance of Na_x_Ca_y_CoO_2_ (0.45 ≤ x ≤ 0.64, 0.02 ≤ y ≤ 0.10) to understand the impact of Ca doping on cathode materials. The ionic radius of Ca^2+^ (1.00 A°) is considerably larger than that of Co^3+^ (0.61 A° for high spin) or Co^4+^ (0.53 A° for high spin), making it unlikely to replace Co directly within the CoO_6_ octahedra. Instead, Ca^2+^ ions are positioned between CoO_6_ slabs, where they strengthen the electrostatic interactions between these layers. While Ca doping reduces the initial discharge capacity, it significantly enhances the structural stability of Na_0.73_CoO_2_. Among the compositions studied, the Na_0.60_Ca_0.07_CoO_2_ cathode exhibited the highest capacity retention over extended cycling (Figure 23f) and superior rate performance (Figure 23g), particularly under high-current conditions.

**Figure 23 micromachines-16-00137-f023:**
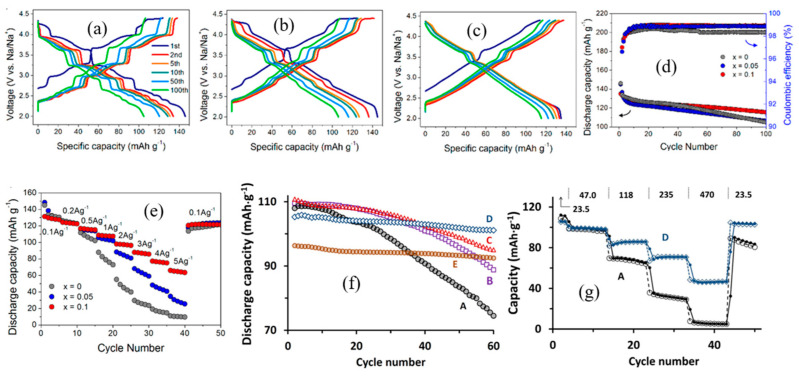
(**a**–**c**) Electrochemical cycling performance of Na/Na_0.67_Co_1-x_Ti_x_O_2_ cells with varying titanium substitution levels (x = 0, x = 0.05, and x = 0.1), illustrating specific discharge capacities as a function of cycle number, (**d**) specific capacity and coulombic efficiency over cycles for each composition, highlighting improved capacity retention and efficiency with titanium doping, (**e**) rate performance comparison of Na/Na_0.67_Co_1-x_Ti_x_O_2_ cells, emphasizing the enhanced rate capability with x = 0.05 and x = 0.1 substitutions. Reprinted from Ref. [134] with submission from the American Chemical Society, (**f**) discharge capacity retention of Na_0.73_CoO_2_ and calcium-doped variants Na_0.68_Ca_0.02_CoO_2_, Na_0.63_Ca_0.05_CoO_2_, Na_0.60_Ca_0.07_CoO_2_, and Na_0.52_Ca_0.10_CoO_2_ at 23.5 mA g ^−1^, demonstrating capacity stabilization with optimized calcium doping levels, and (**g**) discharge (open markers) and charge (solid markers) capacities of Na_0.73_CoO_2_ (A) and Na_0.60_Ca_0.07_CoO_2_ (D) at varying current densities, where marked current values denote performance trends across rate capabilities. Reprinted from Ref. [136] with permission from Elsevier.

Table 3 provides a summary of the capacities achieved with different phases of sodium cobalt oxide. These materials were all examined using non-aqueous electrolytes, such as organic or ionic solvents, with a sodium metal anode serving as the counter electrode.

### 2.6. NaVO_2_

Among the various reported Na_x_VO_2_ compounds, P2- and O3-type compositions have been thoroughly studied. The O3-type Na_x_VO_2_, obtained by reduction in NaVO_3_ under H_2_ atmosphere at 700 °C, demonstrates the highest initial discharge capacity (~140 mAh g^−1^) with full reversibility between Na_0.5_VO_2_ and Na_1_VO_2_ compositions, but suffers from poor cycling stability due to significant slab gliding during the charge–discharge process. The intermediate voltage window (1.6–2.6 V), although indicative of decent electrochemical performance, makes it less attractive for practical applications. Moreover, the O3 phase is sensitive to air, where oxygen slightly oxidizes the material, forming a small amount of an oxidized phase on the particle surface following the reaction:NaVO_2_ + (1 − x)/4 O_2_ → Na_x_VO_2_ + (1 − x)/2 Na_2_O (10)

Figure 24a,b illustrate the electrochemical performance of O3- and P2-Na_x_VO_2_ phases against Na^+^/Na. Figure 24a highlights a highly reversible Na insertion and de-insertion process between 1.2 and 2.4 V. The first charge curve shows a potential plateau indicative of a biphasic domain close to the Na_2/3_VO_2_ composition, followed by multiple potential plateaus associated with more complex behavior. Near the Na_1/2_VO_2_ composition, another single-phase potential step is observed. Figure 24b expands on this by showcasing the extended cycling behavior and capacity retention, underscoring the material’s electrochemical stability and phase transformation reversibility over successive cycles [141,142,143].

Despite these merits, the cell polarization remains significant, as shown in Figure 24c, even at low rates (C/100). Extending the voltage range beyond 2.5 V results in irreversible transitions, as only a small amount of Na can be reinserted (Figure 24d). The discharge curve shape changes significantly, similar to Na_x_TiO_2_ systems, attributed to Ti migration into Na layers. The structural changes are evidenced in Figure 24e, where XRPD patterns for the Na_2/3_VO_2_ and Na_1/2_VO_2_ phases are compared to the pristine NaVO_2_ phase.

The XRPD analysis reveals that for Na_2/3_VO_2_ (2/3 < x < 1), a mixture of starting O3-NaVO_2_ and Na_2/3_VO_2_ phases is formed. For smaller x values, Na_x_VO_2_ phases (1/2 < x < 2/3) crystallize in the C2/m space group, representing a distortion of the trigonal lattice in O3-NaVO_2_. The irreversible decrease in interslab distance suggests vanadium migration into vacancies caused by Na de-intercalation.

The P2-phase Na_x_VO_2_, while challenging to synthesize in a pure single-phase form, emerges as a more promising candidate. Its cycling behavior within the voltage range of 1.4–2.5 V (Figure 24b) shows minimal polarization and a reversible capacity of ~120 mAh g^−1^. However, at higher deintercalation voltages (V > 2.5 V), irreversible phase changes occur. The voltage–composition curve resembles the P2-Na_x_CoO_2_ system (Figure 24f), showing multiple potential plateaus (biphasic domains) and continuous voltage variations (solid-solution phases). Specific potential drops correlate with ordered Na/V ratios (1/2, 5/8, and 2/3), pointing to ordered Na ion arrangements and/or V^3+^/V^4+^ within the VO_2_ slabs [144,145,146,147].

**Figure 24 micromachines-16-00137-f024:**
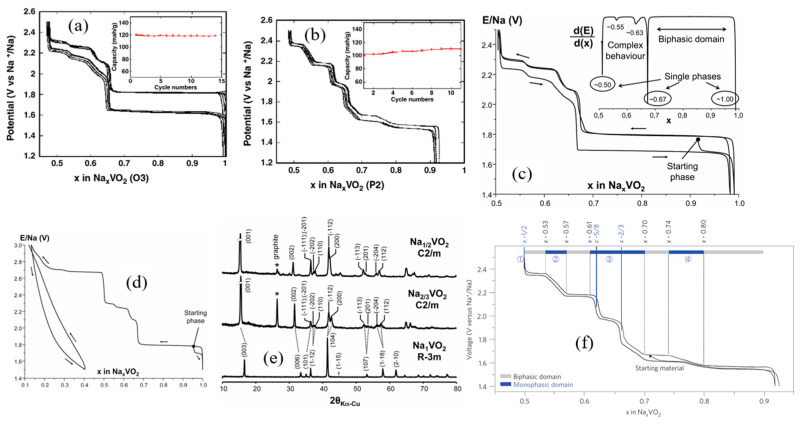
(**a**) Voltage–composition curves for NaVO_2_ (O3), highlighting the reversible formation of Na_1/2_VO_2_ and Na_2/3_VO_2_ during Na de-intercalation and the onset of an electrochemically inactive phase at potentials above 2.4 V, (**b**) voltage–composition curves for Na_0.7_VO_2_ (P2), demonstrating its electrochemical behavior in the sodium-ion battery system. The inset figures show the delivered discharge capacity during limited cycling. Reprinted from Ref. [145] with permission from Elsevier, (**c**) electrochemical cycling of Na_x_VO_2_ (x = 0.92, as determined by lever rule) in the range 1.4–2.5 V at rate C/100. The differential curve dE/dx of the first charge is shown in the inset. Na_x_VO_2_ is a mixture of NaVO_2_ and Na_2/3_VO_2_, (**d**) electrochemical cycling of Na_x_VO_2_ (x = 0.95, as determined by lever rule) in the range 1.5–3.0 V at rate C/100, (**e**) X-ray powder diffractograms of the as-synthesized Na_1_VO_2_ phase and the electrochemically obtained Na_2/3_VO_2_ and Na_1/2_VO_2_ phases. Indexations in the corresponding space groups are shown. (*) is related to graphite peak in the XRD pattern. Reprinted from Ref. [146] with permission from IOP Publishing, (**f**) evolution of cell voltage as a function of sodium content in Na_x_VO_2_ over the 0.5 ≤ x ≤ 0.92 range. The limits of the biphasic domains and the solid solutions are shown by dashed lines. The three single phases for x = 1/2, 5/8, and 2/3 are emphasized by thick blue lines. The numbers 2, 3, and 4 indicate monophasic domains during sodium (de)intercalation. Reprinted from Ref. [144] with permission from Nature.

### 2.7. NaMnO_2_

NaMnO_2_ is an intriguing sub-class of lithium transition metal oxides based on manganese, notable for its high theoretical capacity, environmental friendliness, and ease of synthesis. Na_x_MnO_2_ has been studied since the 1970s, offering a variety of stoichiometries and structures. Depending on sodium content, these materials can form double or triple chains creating 3D tunnel structures (Na/Mn < 0.5) or 2D layered structures (Na/Mn > 0.5). NaMnO_2_ exists in two main phases: at low temperatures, α-NaMnO_2_ adopts a distorted O3 layered structure with monoclinic distortion, attributed to the Jahn–Teller effect of Mn^3+^ ions and at high temperatures, it transforms into orthorhombic β-NaMnO_2_, a layered structure composed of MnO_2_ sheets made of edge-sharing MnO_6_ octahedra, where sodium ions occupy octahedral sites between adjacent MnO_2_ sheets. Density Functional Theory (DFT) calculations estimate similar formation energies for these phases, making it experimentally challenging to synthesize a pure β-NaMnO_2_ phase [148].

The charge–discharge characteristics of a non-aqueous Na/α-NaMnO_2_ cell at varying rates, obtained by solid-state reaction of Na_2_CO_3_ and Mn_2_O_3_ at 700 °C in air, are shown in Figure 25a,b. Almost 0.9 Na per formula unit is extracted during the first charge, delivering an initial charge capacity of 215 mAh g^−1^. Monoclinic NaMnO_2_ provides discharge capacities of 185 and 194 mAh g^−1^ within a voltage range of 2–3.8 V, achieving coulombic efficiencies of 86.4% and 92.8% at C/20 and C/30 rates, respectively (1 C = 244 mAh g^−1^). Coulombic efficiency decreases in subsequent cycles due to side reactions, likely involving the electrolyte, particularly at higher voltages. Lower efficiencies at C/30 suggest prolonged exposure to high voltage exacerbates parasitic reactions or degradation [149]. Overcharging beyond 3.8 V (up to 4.2 V at C/10) shows an additional charge capacity of ~90 mAh g^−1^ between 3.8 V and 4.2 V. However, this additional capacity is not recovered during discharge, indicating it arises from electrolyte decomposition rather than reversible Na-ion intercalation (Figure 25c). Monoclinic NaMnO_2_ exhibits stability issues, as shown in Figure 25d. Structural changes during sodium extraction from O′3-NaMnO_2_ generate internal stress, leading to capacity fading. This fading is linked to bulk structural changes and crystallite size rather than surface reactions. Gradual and irreversible changes in the charge–discharge profiles during cycling reflect the impact of bulk structural evolution [150].

Kubota et al. [150] investigated the phase transitions occurring in monoclinic-NaMnO_2_ during charging. They identified seven highly crystalline O′3-type phases (O′3(1) to O′3(7)) and less crystalline O′3-O1-like phases. The phase evolution proceeds as follows: In the first plateau around 2.6 V, O′3(2) gradually increases at the expense of O′3(1), undergoing a complete phase transition from O′3(1) to O′3(2). A similar two-phase transition process continues with progressive charging up to 3.8 V, involving seven O′3 phases and one O′3-O1 phase.

With progressive sodium extraction, electrostatic repulsion between adjacent MnO_2_ slabs increases, leading to an enlargement of the interslab distance. This trend persists throughout the desodiation process until the emergence of the O′3(7) phase. However, during the transition from O′3(7) to O′3-O1(8) at a voltage plateau around 3.6 V, the interslab distance decreases from 5.63 A° to 5.15 A° (Figure 25e). This transition, associated with decreased crystallinity and high Mn ion migration into the interslab space to form a modulated phase and/or the coexistence of differently oriented domains (Figure 25f), results in the formation of low-crystallinity O′3 phases, causing significant capacity degradation during cycling in a wider voltage range (above 3.55 V).

β-NaMnO_2_ exhibits a distinctive layered structure that sets it apart from the typical polymorphs of α-NaMnO_2_. Unlike the planar MnO_6_ octahedral layers in α-types, which adjust their stacking sequences to form polymorphs like O3, P2, or P3, β-NaMnO_2_ is characterized by zigzag layers of edge-sharing MnO_6_ octahedra. In this structure, Na^+^ ions occupy the octahedral sites between the zigzag MnO_6_ layers (Figure 26a). The electrochemical performance of β-NaMnO_2_, synthesized by firing a mixture of Mn_2_O_3_ and Na_2_CO_3_ first at 950 °C for 24 h, after a temperature ramp of 1 °C/min, and second at 950 °C for 24 h, after ramping the temperature at a faster heating rate of 5 °C/min in oxygen atmosphere, is demonstrated through the load curves in Figure 26b, showing Na extraction and reinsertion processes. During the first charge, a distinct plateau appears between the fully sodiated NaMnO_2_ phase and a desodiated phase close to Na_0.57_MnO_2_, with smaller steps observed at compositions around Na_0.49_MnO_2_ and Na_0.39_MnO_2_. This initial cycle experiences an irreversible capacity loss of approximately 25 mAh g^−1^ (equivalent to 0.1 Na per formula unit). Despite this initial loss, the load curves remain relatively stable over subsequent cycles, with only a slight reduction in capacity. A plateau at 2.7 V corresponds to a phase transition between Jahn–Teller distorted and undistorted structures, accompanied by minimal polarization (below 150 mV). The rate performance of β-NaMnO_2_ at varying charge–discharge rates (Figure 26c) highlights a gradual but irreversible capacity fade over time. Notably, the material does not recover its initial capacity, indicating some structural degradation during cycling, particularly under higher discharge rates (Figure 26d). These findings suggest that while β-NaMnO_2_ demonstrates interesting structural and electrochemical features, its cycling stability requires further investigation and optimization [151].

**Figure 26 micromachines-16-00137-f026:**
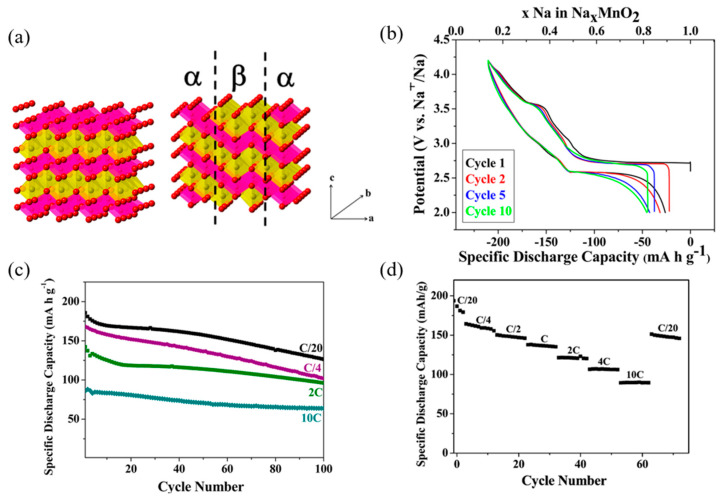
(**a**) Schematic representation of the β-NaMnO_2_ structure in the Pmnm space group. MnO_6_ octahedra are depicted in pink, NaO_6_ octahedra in yellow, and oxygen atoms in red. An intergrowth model on the right illustrates the structural relationship between α- and β-NaMnO_2_. Reprinted from Refs. [151,152] with permission from the American Chemical Society, (**b**) load curves for β-NaMnO_2_ during Na extraction/reinsertion cycles at a C/20 rate (10 mA g^−1^), showing the first (black), second (red), fifth (blue), and tenth (green) cycles, (**c**) cycling performance of β-NaMnO_2_ in the voltage range of 2–4.2 V vs. Na^+^/Na at various rates. Na extraction occurs at C/4, while reinsertion is performed at rates indicated above each dataset. For the C/20 dataset, both extraction and reinsertion occur at C/20, and (**d**) Specific discharge capacities of β-NaMnO_2_ over 100 cycles between 2 and 4.2 V vs. Na^+^/Na at room temperature. The plot illustrates capacity changes with cycling across different rates (C/20 to 10 C). Na extraction occurs at C/4, with reinsertion rates indicated by square markers: black (C/20), purple (C/4), green (C/4), and blue (10 C). Reprinted from Ref. [151] with permission from the American Chemical Society.

To mitigate phase transitions and enhance capacity retention in P2- and O3-type sodium manganese-based layered oxide cathode materials, extensive research has focused on elemental doping and transition metal combinations [107,153,154,155,156,157,158,159,160]. Li et al. [161] synthesized a sodium transition metal oxide, NaMn_1/4_Ni_1/4_Fe_1/4_Co_1/4_O_2_, featuring an equimolar ratio of Ni, Mn, Fe, and Co. This cathode material exhibited an initial discharge capacity of 180 mAh g^−1^, delivered at an average discharge voltage of 3.21 V (Figure 27a). Compared to NaFe_0.5_Co_0.5_O_2_ and NaMn_0.5_Ni_0.5_O_2_, the NaMn_1/4_Ni_1/4_Fe_1/4_Co_1/_4O_2_ cathode demonstrated superior cycling performance across both low and high upper cutoff voltages (Figure 27b,c). The improvements in capacity retention were attributed to the absence of multiple phase transitions (for example, from hexagonal to monoclinic structures) and the reduction in Jahn–Teller distortion of Mn ions.

Further advancements have been made by investigating Fe substitution within the NaMn_1-x_Ni_1-x_Fe_x_O_2_ system. Yuan et al. [162] explored the effect of Fe substitution in NaMn_0.5_Ni_0.5_O_2_ cathode materials, finding that Fe doping successfully suppressed capacity decay during charge–discharge cycling. This enhancement was linked to improved structural stability, specifically the inhibition of MO_2_ layer gliding and phase conversion.

Additionally, Li et al. [163] developed a MoS_2_ coating for NaMn_0.4_Ni_0.4_Fe_0.2_O_2_ particles and studied its impact on the material’s electrochemical performance. The coating not only improved the material’s electrical conductivity but also enhanced its structural stability, even in air. After exposure to air for 10 days, the cathode retained a specific discharge capacity of 128.9 mAh g^−1^ (initially 148.4 mAh g^−1^ at 0.1 C within a 2–4.2 V voltage window). These improvements were attributed to the increased conductivity and reduced side reactions occurring in the electrolyte.

**Figure 27 micromachines-16-00137-f027:**
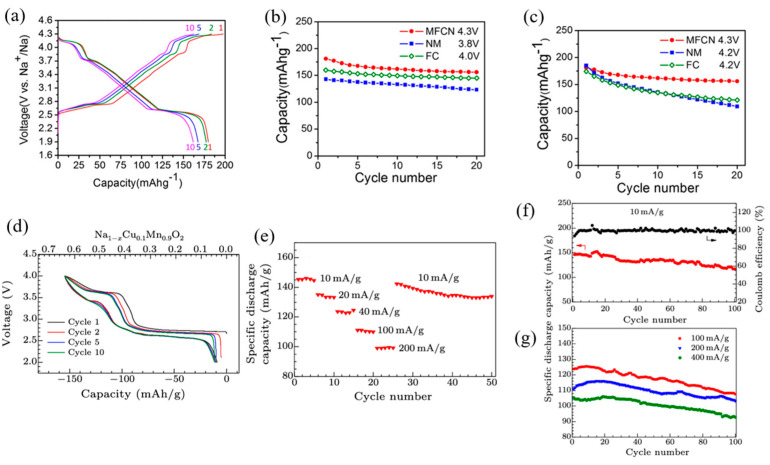
(**a**) Galvanostatic charge and discharge profiles at the 1st, 2nd, 5th, and 10th cycles for NaMn_1/4_Fe_1/4_Co_1/4_Ni_1/4_O_2_ (MFCN), (**b**) discharge capacity of MFCN cycled between 1.9 and 4.3 V at different cycles compared with NaMn_0.5_Ni_0.5_O_2_ (NM) cycled between 1.9 and 3.8 V and NaFe_0.5_Co_0.5_O_2_ (FC) cycled between 1.9 and 4.0 V, all at a C/10 rate. (**c**) Discharge capacity of MFCN cycled between 1.9 and 4.3 V at different cycles compared with NM and FC cycled between 1.9 and 4.2 V, all at a C/10 rate. Reprinted from Ref. [161] with permission from Elsevier, (**d**) charge–discharge curves for NaCu_0.1_Mn_0.9_O_2_ at a rate of C/20 (10 mA/g) in the voltage range of 2.0–4.0 V versus Na^+^/Na. The 1st, 2nd, 5th, and 10th Na extraction/reinsertion cycles are represented in black, red, blue, and green, respectively, (**e**) rate capability of the NaCu_0.1_Mn_0.9_O_2_ electrode material, (**f**) the capacity and coulombic efficiency versus cycle number at 10 mA g^−1^ for NaCu_0.1_Mn_0.9_O_2_, (**g**) the capacity versus cycle number at 100 mA g^−1^, 200 mA g^−1^, and 400 mA g^−1^ for NaCu_0.1_Mn_0.9_O_2_. Reprinted from Ref. [164] with permission from IOP Publishing.

While significant research has been conducted on P2- and O3-type Na_x_MnO_2_ cathode materials, making them strong candidates for high capacity and stability, relatively few studies have addressed modifications of β-Na_x_MnO_2_. For instance, Jiang et al. [164] investigated the effects of Cu doping on the electrochemical performance of β-NaMnO_2_, specifically examining NaCu_0.1_Mn_0.9_O_2_. The charge–discharge curves, capacity retention, and rate performance are presented in Figure 27d–g. As shown in Figure 27d, NaCu_0.1_Mn_0.9_O_2_ exhibited an initial discharge capacity corresponding to 0.6 Na reversible intercalation/deintercalation with a coulombic efficiency of 94.1%, surpassing that of undoped β-NaMnO_2_. Remarkably, NaCu_0.1_Mn_0.9_O_2_ demonstrated excellent reversibility at the voltage plateau of ~3.5 V, unlike the irreversible behavior observed in β-NaMnO_2_.

Moreover, NaCu_0.1_Mn_0.9_O_2_ exhibited improved cycling stability, retaining 80.1% of its initial capacity after 100 cycles, compared to 68.4% retention in β-NaMnO_2_ (Figure 27f). It also showed superior rate performance (Figure 27e,g). This enhanced performance may stem from the absence of local structural rearrangements associated with planar defects in β-NaMnO_2_, which are mitigated by Cu doping.

Based on the discussion of transition metal oxide layered materials for SIBs, it is evident that Mn- and Ni-based compounds are excellent candidates for cathode materials. Wang et al. [116] investigated the substitution of both Li and Ti for Mn and Ni in the structure of NaMn_0.5_Ni_0.5_O_2_. The electrochemical performance of NaLi_1/9_Ni_1/3_Mn_4/9_Ti_1/9_O_2_ and NaMn_0.5_Ni_0.5_O_2_ is presented in Figure 28a,b.

Unlike the charge–discharge curves of NaMn_0.5_Ni_0.5_O_2_, which exhibit multiple voltage plateaus due to complex phase transitions—identified as the primary cause of capacity decay during cycling—the curves for NaLi_1/9_Ni_1/3_Mn_4/9_Ti_1/9_O_2_ are much smoother. This indicates reduced phase transitions, enabling more stable material operation.

Elemental substitution significantly improved the cycling stability of NaMn_0.5_Ni_0.5_O_2_. After 100 cycles at 1 C, the capacity retention increased from 46.97% for the pristine material to 80% for the substituted material, as shown in Figure 28c. The primary reason for this improvement is the expanded Na layer spacing, which enhances Na-ion diffusion and effectively suppresses complex phase transitions (Figure 28d,e).

Beyond conventional elemental doping strategies, structural modifications such as single crystalline and binary structures have been explored as an effective approach to improve transition metal layered oxide cathode materials [165,166]. Single-crystal structural synthesis has emerged as a promising solution to enhance capacity retention during cycling [167,168,169,170,171].

Darga et al. [113] reported that single-crystalline NaMn_0.5_Ni_0.5_O_2_ demonstrated significantly higher capacity retention compared to its polycrystalline counterpart, with 69% retention versus 35% after 200 cycles at 0.2C. The superior stability of the single-crystalline structure is attributed to its enhanced mechanical stability, as illustrated in Figure 28f,g.

As shown, the polycrystalline cathode particles experience pronounced microcracking due to volume expansion, exposing newly formed surfaces of primary particles to parasitic reactions with the electrolyte. This accelerates capacity fading. In contrast, the monocrystalline structured sample exhibited reduced cracking and minimal intragranular fractures, effectively mitigating these degradation mechanisms.

Compared to single-phase materials, heterostructured materials can leverage the multifunctional advantages of multiple phase structures. This design minimizes phase transformation in the material while simultaneously delivering exceptional electrochemical properties [172,173,174,175,176,177].

**Figure 28 micromachines-16-00137-f028:**
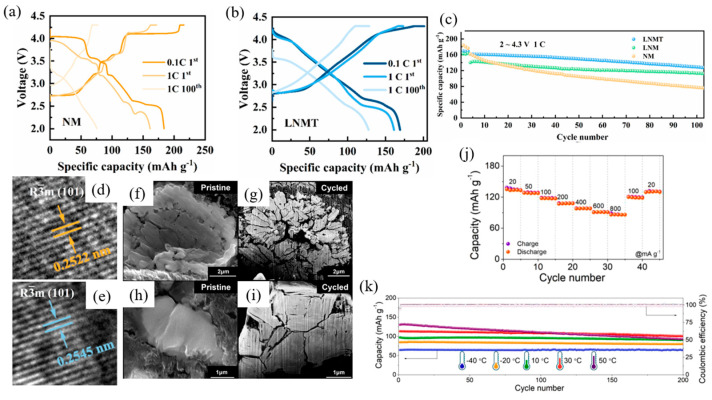
Charge–discharge curves of (**a**) NaMn_0.5_Ni_0.5_O_2_ and (**b**) NaLi_1/9_Ni_1/3_Mn_4/9_Ti_1/9_O_2_ during cycling, (**c**) cycling performance of NaMn_0.5_Ni_0.5_O_2_, NaLi_1/9_Ni_1/3_Mn_5/9_O_2_, and NaLi_1/9_Ni_1/3_Mn_4/9_Ti_1/9_O_2_ cathodes over 100 cycles at 1 C in the voltage range of 2.0–4.3 V. Lattice fringes of (**d**) NaMn_0.5_Ni_0.5_O_2_ and (**e**) NaLi_1/9_Ni_1/3_Mn_4/9_Ti_1/9_O_2_. Reprinted from Ref. [116] with permission from the American Chemical Society. Cross-sectional SEM image of (**f**) pristine polycrystalline particles dispersed in carbon/PVdF and (**g**) FIB-SEM image of polycrystalline particles after 200 cycles. (**h**) Cross-sectional SEM image of a pristine single-crystalline particle and (**i**) FIB-SEM image of single-crystalline particles after 200 cycles. Reprinted from Ref. [113] with permission from the American Chemical Society. (**j**) Rate capability, and (**k**) cycling performance of P2/O3-Na_0.7_Mn_0.4_Ni_0.3_Cu_0.1_Fe_0.1_Ti_0.1_O_1.9_5F_0.1_ at different temperatures. Reprinted from Ref. [178] with permission from Elsevier.

Zhou et al. [178] reported a high-entropy biphasic P2/O3 material with a phase ratio of 23:77, Na_0.7_Mn_0.4_Ni_0.3_Cu_0.1_Fe_0.1_Ti_0.1_O_1.9_5F_0.1_, as an excellent cathode material exhibiting high rate capability and cycling performance (Figure 28j,k) across a wide temperature range of −40 to 50 °C and at high current densities.

The key factor in optimizing the performance of biphasic materials is achieving an ideal phase ratio. An imbalanced ratio can degrade the cycling stability of the multiphase structure, making it less stable than its single-phase counterparts, especially during prolonged cycling.

## 3. Conclusions and Perspectives

Sodium-ion batteries (SIBs) are gaining significant attention as a promising alternative to lithium-ion batteries (LIBs), particularly for large-scale applications. The primary focus has been on improving the cathode materials to enhance their capacity density, operating voltage, and overall energy density. Layered oxide transition metal compounds, widely used as cathode materials in LIBs, have emerged as strong candidates for SIBs.

This review has outlined the key challenges that hinder the widespread application of layered oxide cathode materials in SIBs. The most significant obstacle is the structural instability of these materials, which leads to multiple phase transitions that degrade capacity retention over cycling and under high current conditions. Understanding these phase transitions, especially at higher cutoff voltages, is crucial, as the transition behavior in SIBs differs from that of LIBs due to the distinct sizes of Na and Li ions. Consequently, different phase transition mechanisms are at play, affecting the long-term performance of the cathodes.

As layered oxide materials for SIBs are still in the early stages of development, it is crucial to investigate the side reactions occurring at the cathode–electrolyte interface. Furthermore, detailed research into the thermal stability of battery cells, particularly at elevated temperatures, is essential to ensure their reliability under real-world operating conditions.

Based on the materials discussed, layered oxide compounds containing Mn, Fe, and Ni show great promise as cathode materials for SIBs. These materials can be further optimized through strategies such as introducing electrochemically inactive elements like Nb, Al, and Ti, as well as structural modifications, including single-crystalline designs and biphasic structures combining P2 and O3 phases.

A key focus for new cathode materials is reducing production costs and eliminating the need for expensive elements like Co in their composition. Additionally, the development of coatings is necessary to enhance material conductivity during cycling and high-rate operation, as well as to mitigate parasitic reactions with the electrolyte.

Thus, continued research is imperative to advance the understanding of capacity degradation mechanisms and to refine modification techniques. Ultimately, further exploration into these aspects is critical to enabling the widespread adoption of layered oxide materials as cathodes in SIBs, especially for large-scale energy storage systems.

## Figures and Tables

**Figure 1 micromachines-16-00137-f001:**
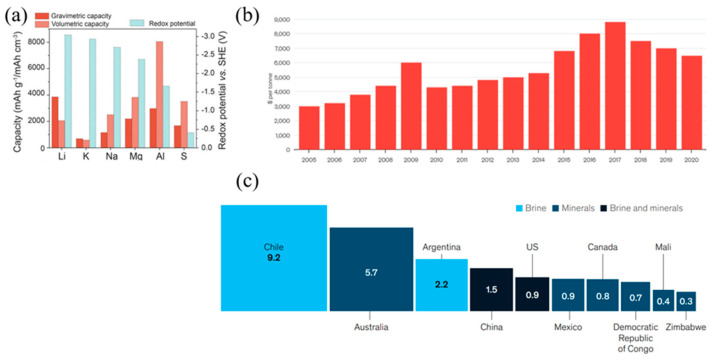
(**a**) Comparison of redox potentials and gravimetric/volumetric capacities for Li, K, Na, Mg, Al, and S. Reprinted from Ref. [9] with permission from Nature, (**b**) price trends of lithium carbonate from 2005 onward. Reprinted from Ref. [10], and (**c**) top 10 countries with the largest lithium reserves, measured in million metric tons. Reprinted from Ref. [11].

**Figure 2 micromachines-16-00137-f002:**
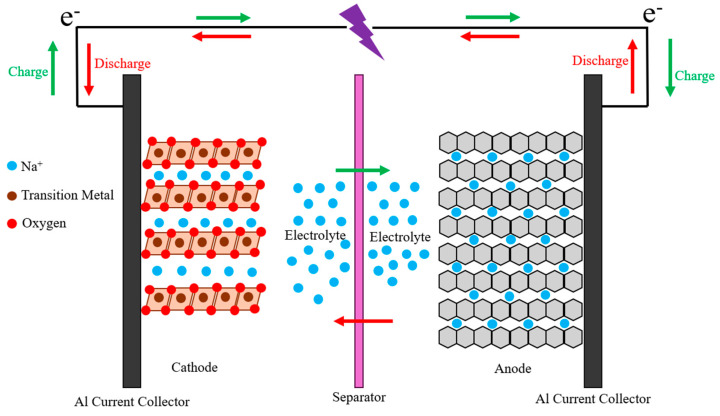
A schematic view of SIB.

**Figure 3 micromachines-16-00137-f003:**
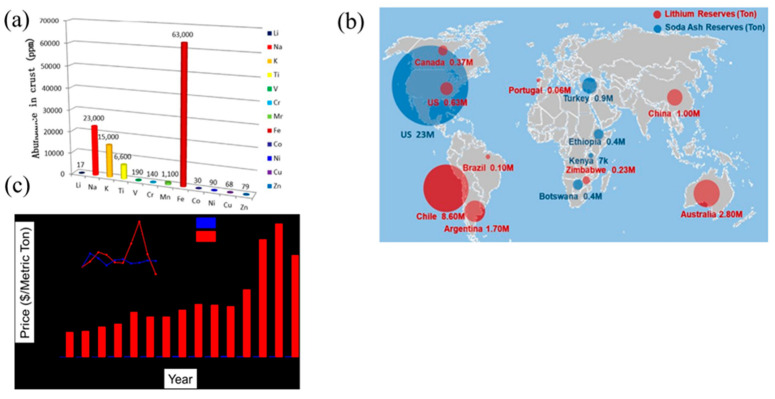
(**a**) Abundance of elements in the Earth’s crust reprinted from Ref. [21] with permission from Elsevier, (**b**) global distribution of lithium and sodium reserves as of 2020, represented by red (lithium) and blue (soda ash) circles, with circle size proportional to reserve quantity in metric tons. Light blue shading indicates oceans as additional sodium sources from brine, and (**c**) price trends for sodium carbonate and lithium carbonate from 2005 to 2019, with an inset highlighting the percentage price changes of both materials over the past decade. Reprinted from Ref. [22] with permission from Wiley.

**Figure 5 micromachines-16-00137-f005:**
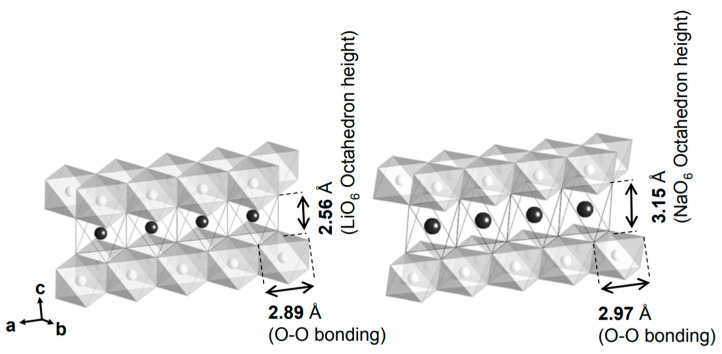
Structural comparison of layered LiCrO_2_ (**left**) and NaCrO_2_ (**right**) crystal models. The illustration highlights the differences in interstitial tetrahedral dimensions and their impact on chromium ion migration during electrochemical cycling. Reprinted from Ref. [48] with permission from IOP Publishing.

**Figure 6 micromachines-16-00137-f006:**
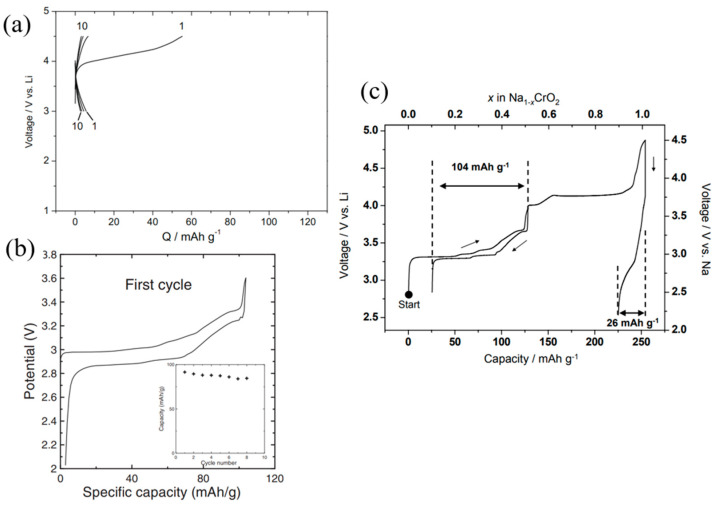
(**a**) Galvanostatic charge and discharge profiles of a Li/LiCrO_2_ cell cycled at a current density of 20 mA g^−1^ within the voltage range of 3.0–4.5 V. Reprinted from Ref. [49] with permission from Elsevier, (**b**) voltage-specific capacity curve of a NaCrO_2_ electrode in a Na/NaCrO_2_ coin cell, cycled between 3.6 and 2.0 V at 25 mA g^−1^. The inset illustrates the specific capacity versus cycle number. Reprinted from Ref. [52] with permission from IOP Publishing, and (**c**) initial charge and discharge profiles of NaCrO_2_ in a Na/NaCrO_2_ cell cycled at 5.0 mA g^−1^. Reprinted from Ref. [48] with permission from IOP Publishing.

**Figure 7 micromachines-16-00137-f007:**
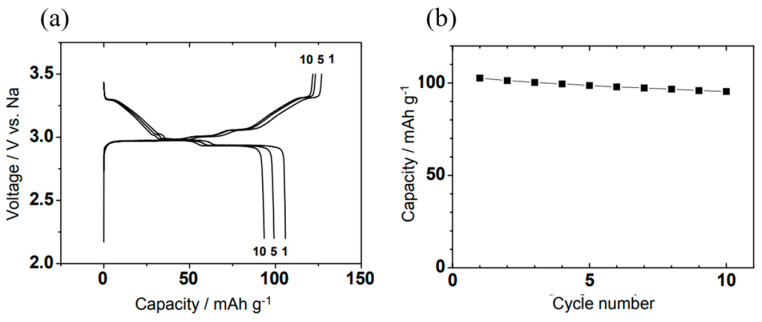
(**a**) Galvanostatic charge and discharge curves of a Na/NaCrO_2_ cell recorded at a current density of 5.0 mA g^−1^ and (**b**) the corresponding variation in discharge capacity over multiple cycles, illustrating the cycling performance of the NaCrO_2_ electrode. Reprinted from Ref. [48] with permission from IOP Publishing.

**Figure 8 micromachines-16-00137-f008:**
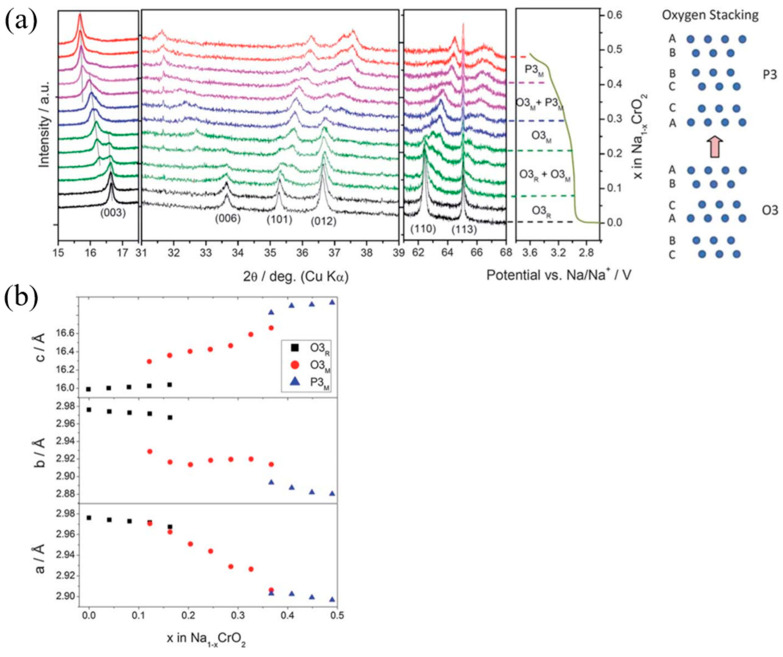
(**a**) In situ X-ray diffraction (XRD) patterns collected during the first charge cycle of a Na/NaCrO_2_ cell up to 3.6 V at a C/12 rate. The displayed 2θ angles were recalculated to the corresponding values for Cu-Kα radiation (λ = 1.54 A°) from the synchrotron XRD wavelength (λ = 0.7747 A°). The voltage–composition profile is shown adjacent to the XRD patterns. On the far right, the schematic oxygen stacking arrangements for O3 and P3 structures are illustrated and (**b**) the evolution of lattice parameters (a, b, and c) as a function of composition during the first charge, derived from in situ XRD data. For clarity, the monoclinic lattice parameters were converted to their rhombohedral equivalents using the transformation equations: a_M_ = √3 a_R_, b_M_ = b_R_, c_M_ = c_R_/3 sinβ. Reprinted from Ref. [51] with permission from the Royal Society of Chemistry.

**Figure 9 micromachines-16-00137-f009:**
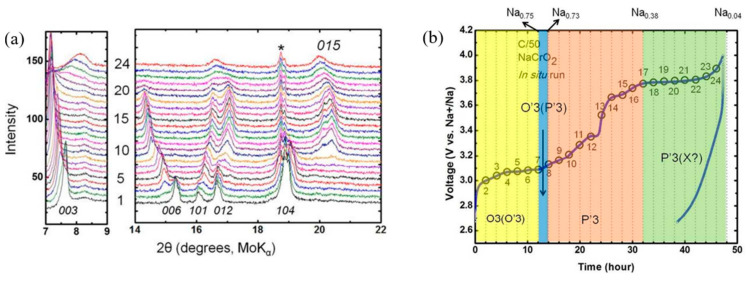
(**a**) In situ XRD patterns collected during the first charge of NaCrO_2_ at a current density of C/50. A peak originating from the Al current collector overlaps with the 104 reflections of NaCrO_2_ at approximately 19° (2θ) and is marked with an asterisk and (**b**) the corresponding electrochemical curve, with scan numbers annotated above it. The graph is divided into four color-shaded regions to emphasize the dominant reactions associated with stacking variations during NaCrO_2_ desodiation. The critical compositions marking transitions between different reaction regimes are also indicated on the upper x-axis. Reprinted from Ref. [53] with permission from the American Chemical Society.

**Figure 10 micromachines-16-00137-f010:**
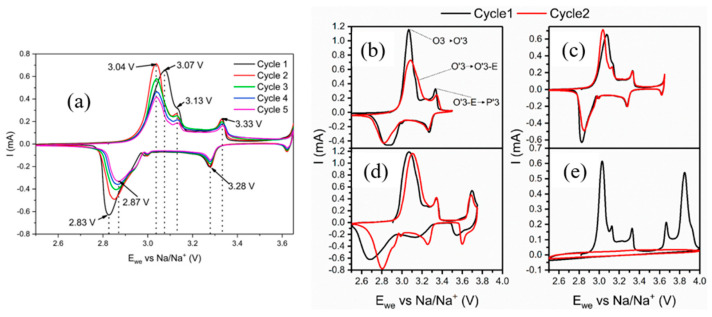
(**a**) Cyclic voltammogram of the first five cycles for the O3-NaCrO_2_ electrode, ranging from 2.5 V to 3.7 V. The cycles are represented in black (cycle 1), red (cycle 2), green (cycle 3), blue (cycle 4), and magenta (cycle 5). Reprinted from Ref. [54] with permission from Elsevier. The cyclic voltammetry data of O3-NaCrO_2_ vs. Na/Na^+^ are shown for different upper potential limits. Two cycles are presented for each potential window, with the first cycle in black and the second in red. The recorded current is plotted against potential. The potential windows used are (**b**) 2.5–3.5 V, (**c**) 2.5–3.65 V, (**d**) 2.5–3.75 V, and (**e**) 2.5–4.0 V. In (**b**), the well-known crystalline-to-crystalline phase transformations (O3 → O′3, O′3 → O′3-E, and O′3-E → P′3) are noted. Reprinted from Ref. [43] with permission from Elsevier.

**Figure 12 micromachines-16-00137-f012:**
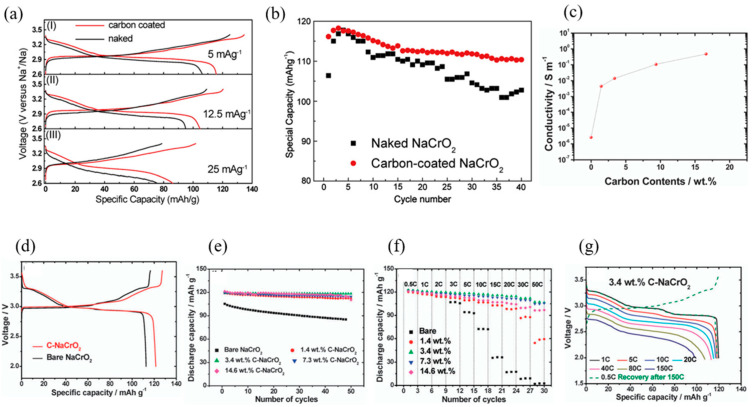
(**a**) Galvanostatic charge–discharge curves of NaCrO_2_/NaClO_4_/Na cells at current densities of 5 mA g^−1^ (I), 12.5 mA g^−1^ (II), and 25 mA g^−1^ (III), (**b**) specific discharge capacity of NaCrO_2_/NaClO_4_/Na cells at a current density of 5 mA g^−1^ as a function of cycle number. Reprinted from Ref. [69] with permission from Elsevier, (**c**) electrical conductivity of NaCrO_2_ as a function of carbon content, (**d**) first charge and discharge curves comparing electrochemical performances of bare and C-NaCrO_2_, using Na metal as the counter electrode and 0.5 M NaPF_6_ in PC:FEC as the electrolyte. Cells were tested in the voltage range of 2.0–3.6 V at a charge-discharge rate of 20 mA g^−1^, (**e**) cycling plots showing capacity as a function of the carbon content, (**f**) rate capability as a function of the carbon content, and (**g**) rate performances of 3.4 wt% C-NaCrO_2_, which delivered the best results among various carbon-coated samples. Reprinted from Ref. [19] with permission from the Royal Society of Chemistry.

**Figure 13 micromachines-16-00137-f013:**
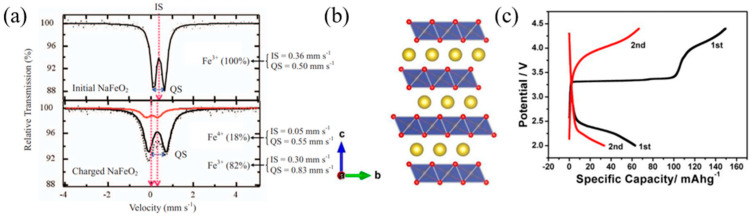
(**a**) Mössbauer spectra of initial and fully charged α-NaFeO_2_ cathodes. The black line represents Fe^3+^, while the red line corresponds to Fe^4+^. Reprinted from Ref. [73] with permission from IOP Publishing, (**b**) schematic illustration of the O3-NaFeO_2_ structure, where Na and Fe ions are octahedrally coordinated with oxygen atoms, forming a layered structure with Na layers between O-Fe-O slabs, and (**c**) voltage profiles of NaFeO_2_ during cycling between 2.0 and 4.4 V versus Na^+^/Na at a current density of 20 mA g^−1^, showing potential plateaus and capacity characteristics. Reprinted from Ref. [31] with permission from Elsevier.

**Figure 14 micromachines-16-00137-f014:**
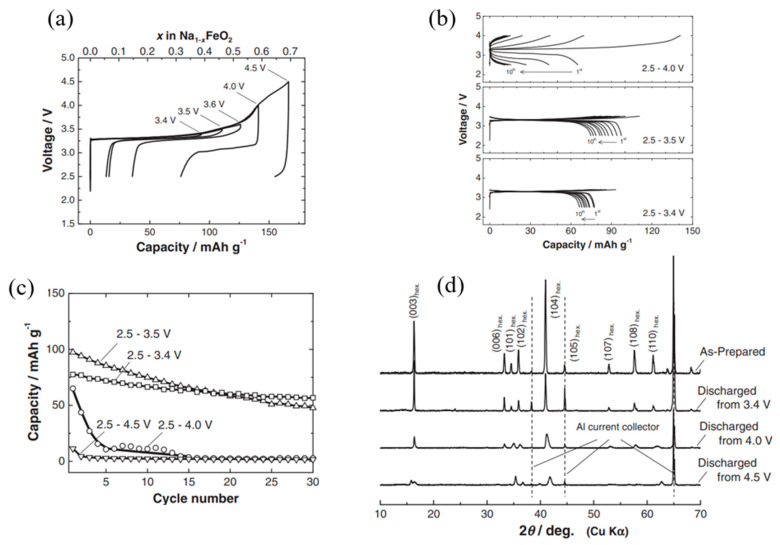
(**a**) Initial charge–discharge profiles of Na/NaFeO_2_ cells at various cutoff voltages, measured at a rate of 12 mA g^−1^, (**b**) charge–discharge curves of Na/NaFeO_2_ cells with different cutoff voltages at the same rate, (**c**) discharge capacity retention over 30 cycles for cells charged to 3.4, 4.0, or 4.5 V and discharged to 2.5 V, and (**d**) XRD patterns of Na_1-x_FeO_2_ composite electrodes before and after the initial charge–discharge cycle at different cutoff voltages. The electrodes were charged (oxidized) to 3.4, 4.0, or 4.5 V and discharged to 2.5 V at a rate of 12.1 mA g^−1^. Reprinted from Ref. [82] with permission from the Electrochemical Society of Japan.

**Figure 15 micromachines-16-00137-f015:**
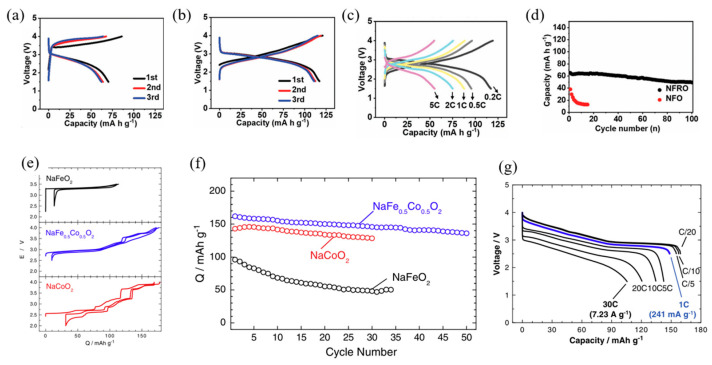
(**a**) Initial charge and discharge curves of Na/NaFeO_2_ cells at a rate of 0.2 C (where 1 C corresponds to 100 mA g^−1^), (**b**) initial charge and discharge curves of Na/Na_4_FeRuO_6_ cells at a rate of 0.2 C, (**c**) rate performance of Na/Na_4_FeRuO_6_ cells at rates of 0.2 C, 0.5 C, 1 C, and 2 C, and (**d**) cycle performance of Na/Na_4_FeRuO_6_ and Na/NaFeO_2_ cells at a rate of 2 C. Reprinted from Ref. [85] with permission from Wiley, (**e**) charge–discharge curves of NaFeO_2_, NaFe_0.5_Co_0.5_O_2_, and NaCoO_2_ in Na cells at a rate of 12 mA g^−1^, (**f**) changes in the discharge capacity over 50 cycles, and (**g**) rate capability of NaFe_0.5_Co_0.5_O_2_. Cells were charged to 4.0 V at a rate of 12 mA g^−1^ and discharged at different rates (C/20 = 12 mA g^−1^ to 30 C = 7260 mA g^−1^). Reprinted from ref. [86] with permission from Elsevier.

**Figure 16 micromachines-16-00137-f016:**
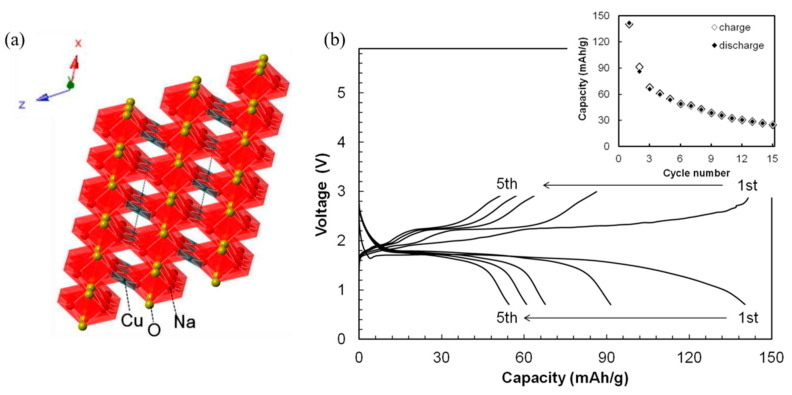
(**a**) Crystal structure of NaCuO_2_ and (**b**) discharge–charge curves of a Na/NaCuO_2_ cell cycled between 0.75 and 3.0 V. The inset illustrates the cycling performance. Reprinted from Ref. [88] with permission from IOP Publishing.

**Figure 17 micromachines-16-00137-f017:**
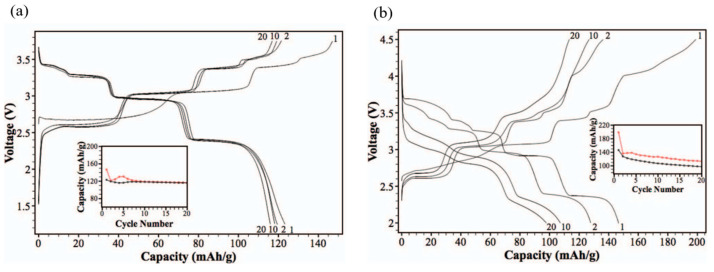
(**a**) Voltage profiles of NaNiO_2_ during multiple cycles at C/10, with the cell galvanostatically cycled within the voltage range of 1.25–3.75 V and (**b**) voltage profiles of NaNiO_2_ cycled between 2.0–4.5 V. Reprinted from Ref. [94] with permission from IOP Publishing.

**Figure 18 micromachines-16-00137-f018:**
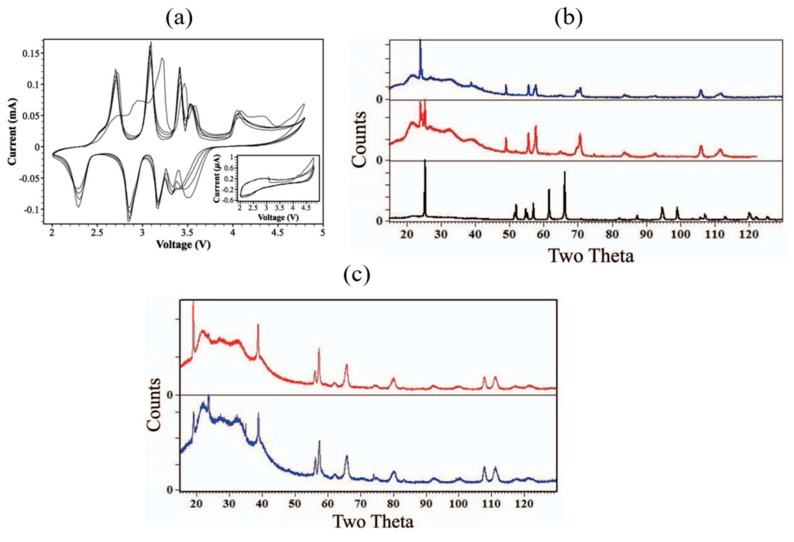
(**a**) Cyclic voltammetry data of NaNiO_2_ (main plot) compared to the electrolyte (inset). The CV data illustrate the oxidation and reduction processes of NaNiO_2_ across multiple cycles, highlighting the reversibility of the material in the 2.0–3.75 V range, (**b**) XRD patterns of pristine NaNiO_2_ (black), NaNiO_2_ mixed with PTFE and carbon black (red), and the cathode material after the first cycle in the 2.0–3.75 V range (blue). The pristine phase corresponds to NaNiO_2_, while the mixed electrode includes an additional Na_0.91_NiO_2_ phase, attributed to minor sodium loss during electrode preparation. The cathode material post-cycling shows only the Na_0.91_NiO_2_ phase, with no evidence of reformation of the fully sodiated NaNiO_2_ phase. Broad peaks between 10° and 30° are due to the Kapton film used during the measurement, and (**c**) XRD patterns of the cathode after charging to 3.75 V (red) and 4.5 V (blue). Both patterns are largely similar, though the cathode charged to 4.5 V exhibits two additional peaks near 23.7° and 35.0°, indicative of the formation of a new phase. The structure of this new phase remains unidentified. The broad peaks between 10° and 30° are also attributed to the Kapton film. The lattice parameters for NaNiO_2_ in this study are reported as a = 2.84 A° and c = 20.8 A°. Reprinted from Ref. [94] with permission from IOP Publishing.

**Figure 19 micromachines-16-00137-f019:**
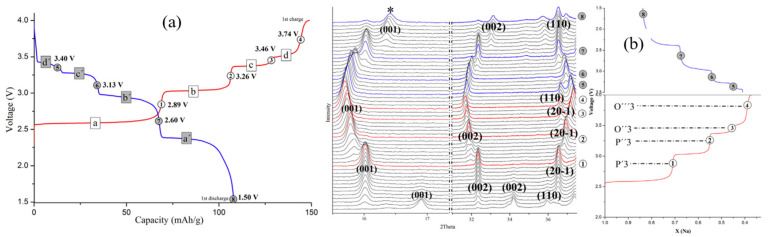
(**a**) First charge–discharge cycle of NaNiO_2_ within the voltage range of 1.5–4.0 V at a C/10 rate. Numbers 1 to 8 indicate the specific voltage stops at which ex situ and relaxed in situ XRD measurements were performed. The charge–discharge process is divided into phases marked by plateaus a to d’ that correspond to key voltage ranges and phase transitions observed during the cycling and (**b**) in situ XRD patterns of the NaNiO_2_ electrode during the first charge–discharge process, showing corresponding phase transitions. The voltage–composition curve, combined with the XRD patterns, highlights the structural changes that occur as sodium is extracted and reinserted, corresponding to the different phases during charge and discharge, as detailed in the charge–discharge plateaus. The phase transitions are tracked in real-time, revealing the changes in the crystal structure at various stages of sodium deintercalation and reintercalation. Reprinted from Ref. [96] with permission from Elsevier.

**Figure 20 micromachines-16-00137-f020:**
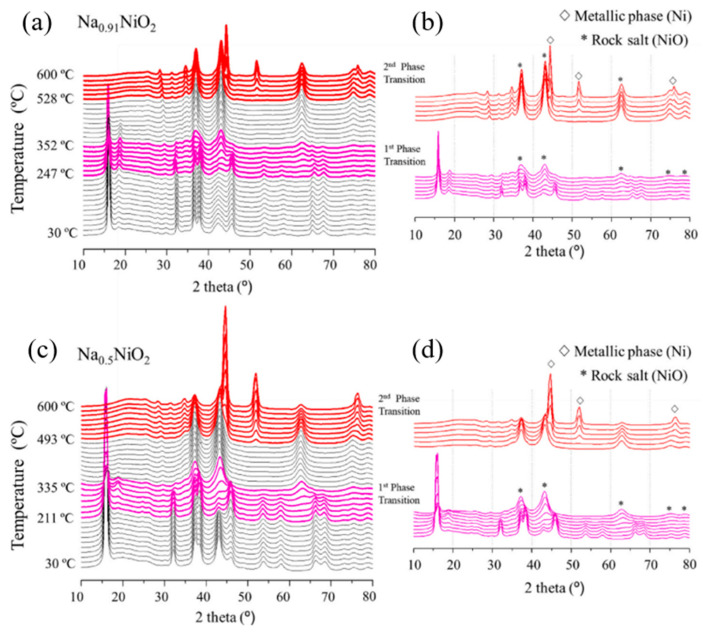
(**a,c**) Time-resolved XRD patterns of Na_0.91_NiO_2_ and Na_0.5_NiO_2_ during heating from 30 to 600 °C, (**b**,**d**) selected regions showing the two phase transition ranges for each material. Reprinted from Ref. [98] with permission from Elsevier.

**Figure 21 micromachines-16-00137-f021:**
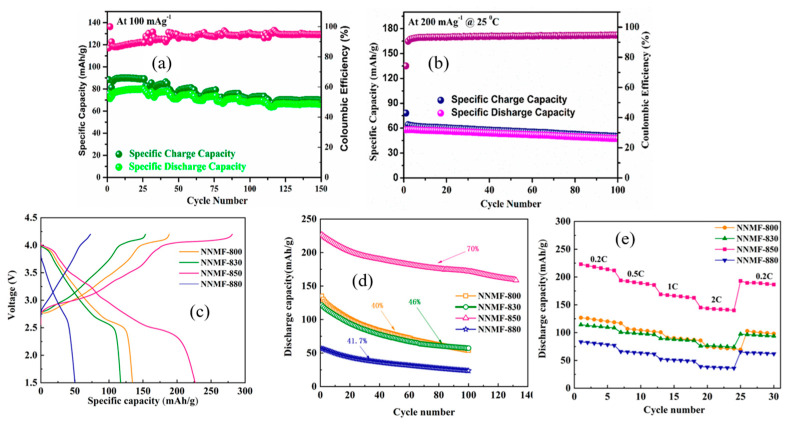
(**a**) Cyclic performance of Na_3_PO_4_@NaNi_0.815_Co_0.15_Al_0.035_O_2_ at a current density of 100 mA g^−1^ at room temperature, (**b**) cyclic performance of Na_3_PO_4_@NaNi_0.815_Co_0.15_Al_0.035_O_2_ at a current density of 200 mA g^−1^ at room temperature. Reprinted from Ref. [119] with permission from the American Chemical Society, (**c**) first charge–discharge curves of NNMF-800, NNMF-830, NNMF-850, and NNMF-880 at a rate of 0.2C (1 C = 160 mA g^−1^), (**d**) capacity retention curves of NNMF-800, NNMF-830, NNMF-850, and NNMF-880 at a rate of 0.2 C, and (**e**) rate cycling plots for NNMF-800, NNMF-830, NNMF-850, and NNMF-880. Reprinted from Ref. [120] with permission from Elsevier.

**Figure 25 micromachines-16-00137-f025:**
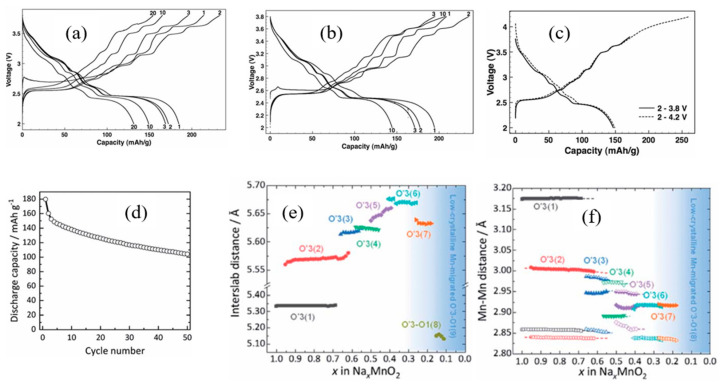
(**a**) Voltage profile of NaMnO_2_ after multiple cycles at a C/20 rate, (**b**) voltage profile of NaMnO_2_ after multiple cycles at a C/30 rate. The cell is galvanostatically cycled between 2.0 V and 3.8 V, (**c**) comparison of the voltage profiles of NaMnO_2_ charged up to 3.8 V and 4.2 V, with the cell cycled at a C/20 rate. Reprinted from Ref. [149] with permission from IOP Publishing, and (**d**) cycling stability of O′3-NaMnO_2_ over 50 cycles at 12 mA g^−1^ in the voltage range of 2.0–3.8 V at 25 °C. Reprinted from Ref. [150] with permission from the Royal Society of Chemistry, (**e**) refinement results exhibiting the changes in interslab distance, and (**f**) in-plane Mn–Mn distance. Reprinted from Ref. [150] with permission from the Royal Society of Chemistry.

**Table 1 micromachines-16-00137-t001:** General comparison between sodium and lithium batteries.

Characteristic	Sodium Batteries	Lithium Batteries
Energy density	150–200 Wh kg^−1^	200–300 Wh kg^−1^
Nominal voltage	3.2–3.3 V	3.6–3.7 V
Environmentally friendly	Can be transported at zero volt	Should be always stored with a minimum charge
Availability	Mostly worldwide available at about 500 times higher the lithium	Availability is limited to some countries and not abundant
Raw material price	~330 U.S. dollar per metric ton for Na_2_CO_3_	~68,000 U.S. dollar per metric ton for Li_2_CO_3_
Lifespan	5–10 years	5–15 years

**Table 2 micromachines-16-00137-t002:** Electrochemical performance of Cr-based and modified Cr-based cathode for SIBs.

Researchers	Pristine Material	Modified Material	Testing Condition	Pristine Material [mAh g^−1^]	Modified Material [mAh g^−1^]
Ikhe et al. [60]	NaCrO_2_	NaCr_0.98_Al_0.02_O_2_@Cr_2_O_3_	2.0–3.6 V, RT * 600 mAh g^−1^	108.5 (initial) 56 (500 cycles)	106.6 (initial) 107 (1000 cycles)
Li et al. [64]	NaCrO_2_	Na_0.95_Cr_0.95_Ti_0.05_O_2_	2.3–3.6 V, RT 120 mAh g^−1^	104 (initial) 52 (800 cycles)	98 (initial) 78 (800 cycles)
Ma et al. [62]	NaCrO_2_	Na_0.9_Cr_0.95_Sn_0.05_O_2_	2.5–3.5 V, RT 600 mAh g^−1^	107 (initial) 50 (1000 cycles)	114 (initial) 81 (1000 cycles)
Zheng et al. [46]	NaCrO_2_	Na_0.9_Ca_0.05_CrO_2_	2–3.6 V, RT 120 mAh g^−1^	110 (initial) 52 (500 cycles)	118 (initial) 90 (500 cycles)
Cai et al. [66]	NaCrO_2_	NaCrO_2_@nitrogen doped carbon	2.0–3.6 V, RT 1200 mAh g^−1^	90 (initial) 47.5 (1000 cycles)	110 (initial) 100.2 (1000 cycles)
Lee et al. [61]	NaCrO_2_	Na_0.9_Ca_0.035_Cr_0.97_Ti_0.03_O_2_	1.5–3.8 V, RT 2400 mAh g^−1^	34 (initial) 22.1 (500 cycles)	76 (initial) 61.7 (500 cycles)
Wu et al. [70]	NaCrO_2_	NaCrO_2_@PVDF-La_2_(CO_3_)_3_·8H_2_O	2.3–3.5 V, RT 240 mAh g^−1^	119.6 (initial) 56.5 (400 cycles)	111.8 (initial) 103.5 (900 cycles)

* RT stands for room temperature.

**Table 3 micromachines-16-00137-t003:** Specific capacities reported for various phases of Na_x_CoO_2_.

Compound	Phase Type	Discharge Specific Capacity (mAh g^−1^)	Cycling Rate
Na_0.67_CoO_2_ [20]	P2	105.0	2 μA cm^−2^
Na_0.74_CoO_2_ [137]	P2	107.0	0.1 C
Na_0.71_CoO_2_ [138]	P2	69.7	0.08 C
Na_0.74_CoO_2_ [135]	P2	134.0	11.1 mA g^−1^
Na_0.67_CoO_2_ [139]	P2	146.8	10 mAg^−1^
Na_0.67_CoO_2_ [134]	P2	146.0	100 mA g^−1^
Na_0.7_CoO_2_ [140]	P2	125.0	5 mA g^−1^
Na_0.99_CoO_2_ [20]	O3	118.0	4.3 μA cm^−2^
NaCoO_2_ [86]	O3	140.0	12 mA g^−1^
Na_0.57_CoO_2_ [124]	P2	68.0	0.7 C (0.1 A g^−1^)

## Data Availability

No new data were created or analyzed in this study. Data sharing is not applicable to this article.

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
