# Peer review of "Review of Layered Transition Metal Oxide Materials for Cathodes in Sodium-Ion Batteries"

_micromachines, 2025, doi:10.3390/mi16020137_

Round 1
Reviewer 1 Report
Comments and Suggestions for Authors
This work provides a systematic discussion of the structural properties, electrochemical performance, degradation mechanisms, and corresponding modification strategies of layered sodium transition metal oxides, offering valuable technical support for the development of SIBs. This study reviews the research progress of NaCrO2, NaFeO2, NaCuO2, NaNiO2, NaCoO2, NaVO2, and NaMnO2, with a focus on the relationship between cathode material composition, phase transitions, and electrochemical performance. Overall, this work is well-structured, logically coherent, and rich in content. Therefore, it is recommended for publication after addressing the following concerns:
1. On page 24, the authors discuss the development of NFM cathodes. It is suggested to include a comparison of the electrochemical performance of unary oxides with binary and ternary oxides to emphasize the impact and significance of compositional variations.
2. As a widely regarded cathode material, it is recommended to expand the discussion on NaNi0.5Mn0.5O2 to provide a more comprehensive perspective. (10.1002/adfm.202414627; 10.1021/acsnano.4c04847; 10.1002/adfm.202301568; 10.1016/j.ensm.2022.02.043; 10.1021/acsami.2c12098)
3. It is recommended to include a comparative analysis of the electrochemical performance of different cathodes, as this would provide readers with a clearer and more intuitive understanding of their relative advantages and limitations.
4. Beyond conventional doping, high-entropy strategies and multiphase structure designs have been shown to significantly enhance electrochemical performance. It is recommended to supplement the discussion with these advanced strategies. (10.1002/adma.202312300; 10.1038/s41560-024-01616-5; 10.1002/adfm.202206154; 10.1021/jacs.2c02353; 10.1002/adma.202401048)
5. It is recommended to update the references with more recent publications, as this would enhance the relevance of the study and underscore the timeliness of the work.
6. In the perspective section, it is suggested to explore potential candidates for commercial cathode materials and propose appropriate modification strategies to further enrich and enhance the content.
Reviewer 2 Report
Comments and Suggestions for Authors
1. The authors should also mention the synthesis method of different types of cathode material for sodium-ion batteries in the manuscript.
2. The structure of the sodium-ion batteries should be provided in the manuscript for better understanding of the working mechanism.
3. The sodium solid state battery is promising in real applications due to their benefits like non-flammable. The authors should also discuss it in the introduction sections.
4. Any ideals of synthesizing new types of layered transition metal oxide cathode materials for sodium ion batteries? The research prospect should be briefly discussed in the manuscript.
5. I will suggest the authors add the plot highlighting the benefits and current limitations of sodium ion batteries compared to lithium ion batteries for better understanding.
6. How is the thermal stability of these layered transition metal oxide cathode materials for sodium ion batteries discussed in the manuscript?
7. Some TEM images of layered transition metal oxide cathode materials before and after battery operations can be provided in the manuscript to help understand the structure evolution and phase transition behaviors.
